# Statistical-Query Lower Bounds via Functional Gradients

**Surbhi Goel**
Microsoft Research
New York, NY, USA
goel.surbhi@microsoft.com

**Aravind Gollakota**
University of Texas at Austin
Austin, TX, USA
aravindg@cs.utexas.edu

**Adam Klivans**
University of Texas at Austin
Austin, TX, USA
klivans@cs.utexas.edu

## Abstract

We give the first statistical-query lower bounds for agnostically learning any non-polynomial activation with respect to Gaussian marginals (e.g., ReLU, sigmoid, sign). For the specific problem of ReLU regression (equivalently, agnostically learning a ReLU), we show that any statistical-query algorithm with tolerance $n^{-(1/\epsilon)^b}$ must use at least $2^{n^c}\epsilon$ queries for some constants $b, c > 0$, where $n$ is the dimension and $\epsilon$ is the accuracy parameter. Our results rule out *general* (as opposed to correlational) SQ learning algorithms, which is unusual for real-valued learning problems. Our techniques involve a gradient boosting procedure for "amplifying" recent lower bounds due to Diakonikolas et al. (COLT 2020) and Goel et al. (ICML 2020) on the SQ dimension of functions computed by two-layer neural networks. The crucial new ingredient is the use of a nonstandard convex functional during the boosting procedure. This also yields a best-possible reduction between two commonly studied models of learning: agnostic learning and probabilistic concepts.

## 1 Introduction

In this paper we continue a recent line of research exploring the computational complexity of fundamental primitives from the theory of deep learning [GKK19, YS19, DKKZ20, YS20, DGK+20, FCG20]. In particular, we consider the problem of fitting a single nonlinear activation to a joint distribution on $\mathbb{R}^n \times \mathbb{R}$. When the nonlinear activation is ReLU, this problem is referred to as ReLU regression or agnostically learning a ReLU. When the nonlinear activation is sign and the labels are Boolean, this problem is equivalent to the well-studied challenge of agnostically learning a halfspace [KKMS08].

We consider arguably the simplest possible setting—when the marginal distribution is Gaussian—and give the first statistical-query lower bounds for learning broad classes of nonlinear activations. The statistical-query model is a well-studied framework for analyzing the sample complexity of learning problems and captures most known learning algorithms. For common activations such as ReLU, sigmoid, and sign, we give complementary upper bounds, showing that our results cannot be significantly improved.

Let $\mathcal{H}$ be a function class on $\mathbb{R}^n$, and let $\mathcal{D}$ be a labeled distribution on $\mathbb{R}^n \times \mathbb{R}$ such that the marginal on $\mathbb{R}^n$ is $D = \mathcal{N}(0, I_n)$. We say that a learner learns $\mathcal{H}$ under $\mathcal{D}$ with error $\epsilon$ if it outputs a function

$f$ such that

$$\mathbb{E}_{(x,y)\sim\mathcal{D}}[f(x)y] \geq \max_{f'\in\mathcal{H}} \mathbb{E}_{(x,y)\sim\mathcal{D}}[f'(x)y] - \epsilon.$$

One can show that this loss captures 0-1 error in the Boolean case, as well as squared loss in the ReLU case whenever the learner is required to output a nontrivial hypothesis (i.e., a hypothesis with norm bounded below by some constant $c > 0$). (See Appendices I and J for details.)

For ReLU regression, we obtain the following exponential lower bound:

**Theorem 1.1.** *Let $\mathcal{H}_{\mathrm{ReLU}}$ be the class of ReLUs on $\mathbb{R}^n$ with unit weight vectors. Suppose that there is an SQ learner capable of learning $\mathcal{H}_{\mathrm{ReLU}}$ under $\mathcal{D}$ with error $\epsilon$ using $q(n, \epsilon, \tau)$ queries of tolerance $\tau$. Then for any $\epsilon$, there exists $\tau = n^{-(1/\epsilon)^b}$ such that $q(n, \epsilon, \tau) \geq 2^{n^c}\epsilon$ for some $0 < b, c < 1/2$. That is, a learner must either use tolerance smaller than $n^{-(1/\epsilon)^b}$ or more than $2^{n^c}\epsilon$ queries.*

Prior work due to Goel et al. [GKK19] gave a quasipolynomial SQ lower bound (with respect to correlational queries) for ReLU regression when the learner is required to output a ReLU as its hypothesis.

For the sigmoid activation we obtain the following lower bound:

**Theorem 1.2.** *Consider the above setup with $\mathcal{H}_\sigma$, the class of unit-weight sigmoid units on $\mathbb{R}^n$. For any $\epsilon$, there exists $\tau = n^{-\Theta(\log^2 1/\epsilon)}$ such that $q(n, \epsilon, \tau) \geq 2^{n^c}\epsilon$ for some $0 < c < 1/2$.*

We are not aware of any prior work on the hardness of agnostically learning a sigmoid with respect to Gaussian marginals.

For the case of halfspaces, a classic result of Kalai et al. [KKMS08] showed that any halfspace can be agnostically learned with respect to Gaussian marginals in time and sample complexity $n^{O(1/\epsilon^4)}$, which was later improved to $n^{O(1/\epsilon^2)}$ [DKN10]. The only known hardness result for this problem is due to Klivans and Kothari [KK14] who gave a quasipolynomial lower bound based on the hardness of learning sparse parity with noise. Here we give the first exponential lower bound:

**Theorem 1.3.** *Consider the above setup with $\mathcal{H}_{\mathsf{hs}}$, the class of unit-weight halfspaces on $\mathbb{R}^n$. For any $\epsilon$, there exists $\tau = n^{-\Theta(1/\epsilon)}$ such that $q(n, \epsilon, \tau) \geq 2^{n^c}\epsilon$ for some fixed constant $0 < c < 1/2$.*

Since it takes $\Theta(1/\tau^2)$ samples to simulate a query of tolerance $\tau$, our constraint on $\tau$ here can be interpreted as saying that to avoid the exponential query lower bound, one needs sample complexity at least $\Theta(1/\tau^2) = n^{\Theta(1/\epsilon)}$, nearly matching the upper bound of [KKMS08, DKN10].

These results are formally stated and proved in Section 5. More generally, we show in Appendix C that our results give superpolynomial SQ lower bounds for agnostically learning any non-polynomial activation. (See Appendix A for some discussion of subtleties in interpreting these bounds.)

A notable property of our lower bounds is that they hold for *general* statistical queries. As noted by several authors [APVZ14, VW19], proving SQ lower bounds for real-valued learning problems often requires further restrictions on the types of queries the learner is allowed to make (e.g., correlational or Lipschitz queries).

Another consequence of our framework is the first SQ lower bound for agnostically learning monomials with respect to Gaussian marginals. In contrast, for the realizable (noiseless) setting, recent work due to Andoni et al. [ADHV19] gave an attribute-efficient SQ algorithm for learning monomials. They left open the problem of making their results noise-tolerant. We show that in the agnostic setting, no efficient SQ algorithm exists; the proof is in Appendix B.

**Theorem 1.4.** *Consider the above setup with $\mathcal{H}_{\mathsf{mon}}$, the class of multilinear monomials of degree at most $d$ on $\mathbb{R}^n$. For any $\epsilon \leq \exp(-\Theta(d))$ and $\tau \leq \epsilon^2$, $q(n, \epsilon, \tau) \geq n^{\Theta(d)}\tau^{5/2}$.*

**Our Approach** Our approach deviates from the standard template for proving SQ lower bounds and may be of independent interest. In almost all prior work, SQ lower bounds are derived by constructing a sufficiently large family of nearly orthogonal functions with respect to the underlying marginal distribution. Instead, we will use a reduction-based approach:

- We show that an algorithm for agnostically learning a single nonlinear activation $\phi$ can be used as a subroutine for learning depth-two neural networks of the form $\psi(\sum_i \phi(w^i \cdot x))$

where $\psi$ is any monotone, Lipschitz activation. This reduction involves an application of functional gradient descent via the Frank–Wolfe method with respect to a (nonstandard) convex surrogate loss.

- We apply recent work due to [DKKZ20] and [GGJ+20] that gives SQ lower bounds for learning depth-two neural networks of the above form in the probabilistic concept model. For technical reasons, our lower bound depends on the norms of these depth-two networks, and we explicitly calculate them for ReLU and sigmoid.

- We prove that the above reduction can be performed using only statistical queries. To do so, we make use of some subtle properties of the surrogate loss and the functional gradient method itself.

Our reduction implies the following new relationship between two well-studied models of learning: if concept class $\mathcal{C}$ is efficiently agnostically learnable, then the class of monotone, Lipschitz functions of linear combinations of $\mathcal{C}$ is learnable in the *probabilistic concept* model due to Kearns and Schapire [KS94]. We cannot hope to further strengthen the conclusion to *agnostic* learnability of monotone, Lipschitz functions of combinations of $\mathcal{C}$: the concept class of literals *is* agnostically learnable, but we show exponential SQ lower bounds for agnostically learning the class of majorities of literals, i.e., halfspaces (see also [KK14]).

**Related Work**    Several recent papers have considered the computational complexity of learning simple neural networks [Bac17, GKKT17, YS20, FCG20, KK14, LSSS14, SVWX17, VW19, GKK19, GGJ+20, DKKZ20]. The above works either consider one-layer neural networks (as opposed to learning single neurons), or make use of discrete distributions (rather than Gaussian marginals), or hold for narrower classes of algorithms (rather than SQ algorithms). Goel et al. [GKK19] give a quasipolynomial correlational SQ lower bound for proper agnostic learning of ReLUs with respect to Gaussian marginals. They additionally give a similar computational lower bound assuming the hardness of learning sparse parity with noise.

The idea of using functional gradient descent to learn one hidden layer neural networks appears in work due to Bach [Bac17], who considered an "incremental conditional gradient algorithm" that at each iteration implicitly requires an agnostic learner to complete a "Frank–Wolfe step." A key idea in our work is to optimize with respect to a particular convex functional (surrogate loss) in order to obtain SQ learnability for depth-two neural networks *with a nonlinear output activation.* We can then leverage SQ lower bounds for this broader class of neural networks.

Functional gradient descent or gradient boosting methods have been used frequently in learning theory, especially in online learning (see e.g., [Fri01, MBBF00, SF12, BHKL15, Haz16].)

For Boolean functions, the idea to use boosting to learn majorities of a base class appeared in Jackson [Jac97], who boosted a weak parity learning algorithm in order to learn thresholds of parities (TOP). Agnostic, distribution-specific boosting algorithms for Boolean functions have appeared in works due to Kalai and Kanade [KK09] and also Feldman [Fel10]. Agnostic boosting in the context of the SQ model is explored in [Fel12], where an SQ lower bound is given for agnostically learning monotone conjunctions with respect to the uniform distribution on the Boolean hypercube.

The SQ lower bounds we obtain for agnostically learning halfspaces can be derived using one of the above boosting algorithms due to Kalai and Kanade [KK09] or Feldman [Fel10] in place of functional gradient descent, as halfspaces are Boolean functions.

**Independent Work**    Independently and concurrently, Diakonikolas et al. [DKZ20] have obtained similar results for agnostically learning halfspaces and ReLUs. Rather than using a reduction-based approach, they construct a hard family of Boolean functions. They show that an agnostic learner for halfspaces or ReLUs would yield a learner for this family, which would solve a hard unsupervised distribution-learning problem considered in [DKS17]. Quantitatively, the lower bound they obtain is that agnostic learning of halfspaces or ReLUs up to excess error $\epsilon$ using queries of tolerance $n^{-\mathrm{poly}(1/\epsilon)}$ requires at least $n^{\mathrm{poly}(1/\epsilon)}$ queries. These results are technically incomparable with ours. For queries of similar tolerance, our bound of $2^{n^c}\epsilon$ scales exponentially with $n$ whereas theirs only scales polynomially, so that for any constant $\epsilon$ our bound is exponentially stronger. But our bound does not scale directly with $1/\epsilon$ (other than via the induced constraint on tolerance, which does scale as $n^{-\mathrm{poly}(1/\epsilon)}$). Our work also extends to general non-polynomial activations, while theirs does not.

**Organization** We cover the essential definitions, models and existing lower bounds that we need in the preliminaries. Our main reduction, which says that if we could agnostically learn a single neuron, then we could learn depth-two neural networks composed of such neurons, is set up as follows. In Section 3 we explain our usage of functional gradient descent, with Assumption 3.1 formally stating the kind of agnostic learning guarantee we require for a single neuron. The main reduction itself is Theorem 4.1, the subject of Section 4. In Section 5 we derive the formal lower bounds which follow as a consequence of our reduction. Finally in Section 6, we contrast these lower bounds by also including some simple upper bounds. The more technical proofs may be found in the appendices.

## 2 Preliminaries

**Notation** Let $D$ be a distribution over $\mathbb{R}^n$, which for us will be the standard Gaussian $\mathcal{N}(0, I_n)$ throughout. We will work with the $L^2$ space $L^2(\mathbb{R}^n, D)$ of functions from $\mathbb{R}^n$ to $\mathbb{R}$, with the inner product given by $\langle f, g \rangle_D = \mathbb{E}_D[fg]$. The corresponding norm is $\|f\|_D = \sqrt{\mathbb{E}_D[f^2]}$. We refer to the ball of radius $R$ as $\mathcal{B}_D(R) = \{f \in L^2(\mathbb{R}^n, D) \mid \|f\|_D \leq R\}$. We omit the subscripts when the meaning is clear from context. Given vectors $u, v \in \mathbb{R}^n$, we will refer to their Euclidean dot product by $u \cdot v$ and the Euclidean norm by $\|u\|_2$. Given a function $\ell(a, b)$ we denote its partial derivative with respect to its first parameter, $\frac{\partial \ell}{\partial a}(a, b)$, by $\partial_1 \ell(a, b)$.

A Boolean probabilistic concept, or $p$-concept, is a function that maps each point $x$ to a random $\{\pm 1\}$-valued label $y$ in such a way that $\mathbb{E}[y|x] = f^*(x)$ for a fixed function $f^* : \mathbb{R}^n \to [-1, 1]$, known as its conditional mean function. We will use $D_{f^*}$ to refer to the (unique) induced labeled distribution on $\mathbb{R}^n \times \{\pm 1\}$, i.e. we say $(x, y) \sim D_{f^*}$ if the marginal distribution of $x$ is $D$ and $\mathbb{E}[y|x] = f^*(x)$. We also sometimes use $y \sim f^*(x)$ to say that $y \in \{\pm 1\}$ and $\mathbb{E}[y|x] = f^*(x)$.

**Statistical Query (SQ) Model** A statistical query is specified by a query function $\phi : \mathbb{R}^n \times \mathbb{R} \to [-1, 1]$. Given a labeled distribution $\mathcal{D}$ on $\mathbb{R}^n \times \mathbb{R}$, the SQ model allows access to an SQ oracle (known as the STAT oracle in the SQ literature) that accepts a query $\phi$ of specified tolerance $\tau$, and responds with a value in $[\mathbb{E}_{(x,y)\sim\mathcal{D}}[\phi(x,y)] - \tau, \mathbb{E}_{(x,y)\sim\mathcal{D}}[\phi(x,y)] + \tau]$.

Let $\mathcal{C}$ be a class of Boolean $p$-concepts over $\mathbb{R}^n$, and let $D$ be a distribution on $\mathbb{R}^n$. We say that a learner learns $\mathcal{C}$ with respect to $D$ up to $L^2$ error $\epsilon$ if, given only SQ oracle access to $D_{f^*}$ for some unknown $f^* \in \mathcal{C}$, and using arbitrary queries, it is able to output $f : \mathbb{R}^n \to [-1, 1]$ such that $\|f - f^*\|_D \leq \epsilon$. It is worth emphasizing that a query to $D_{f^*}$ takes in a Boolean rather than a real-valued label, i.e. is really of the form $\phi : \mathbb{R}^n \times \{\pm 1\} \to [-1, 1]$. In contrast, a query to a generic distribution $\mathcal{D}$ on $\mathbb{R}^n \times \mathbb{R}$ takes in real-valued labels, and in Assumption 3.1 we define a form of learning that operates in this more generic setting.

One of the chief features of the SQ model is that one can give strong information theoretic lower bounds on learning a class $\mathcal{C}$ in terms of its so-called statistical dimension, which can be thought of as roughly measuring how many highly-orthogonal functions $\mathcal{C}$ contains.

**Definition 2.1.** Let $D$ be a distribution on $\mathbb{R}^n$, and let $\mathcal{C}$ be a real-valued or Boolean concept class on $\mathbb{R}^n$. The *average (un-normalized) correlation* of $\mathcal{C}$ is defined to be $\rho_D(\mathcal{C}) = \frac{1}{|\mathcal{C}|^2} \sum_{c,c' \in \mathcal{C}} |\langle c, c' \rangle_D|$. The *statistical dimension on average* at threshold $\gamma$, $\mathrm{SDA}_D(\mathcal{C}, \gamma)$, is the largest $d$ such that for all $\mathcal{C}' \subseteq \mathcal{C}$ with $|\mathcal{C}'| \geq |\mathcal{C}|/d$, $\rho_D(\mathcal{C}') \leq \gamma$.

In the $p$-concept setting, lower bounds against general queries in terms of SDA were first formally shown in [GGJ+20].

**Theorem 2.2** ([GGJ+20], Cor. 4.6)**.** *Let $D$ be a distribution on $\mathbb{R}^n$, and let $\mathcal{C}$ be a $p$-concept class on $\mathbb{R}^n$. Say our queries are of tolerance $\tau$, the final desired $L^2$ error is $\epsilon$, and that the functions in $\mathcal{C}$ satisfy $\|f^*\| \geq \beta$ for all $f^* \in \mathcal{C}$. For technical reasons, we will require $\tau \leq \epsilon^2$, $\epsilon \leq \beta/3$ (see Appendix A for some discussion). Then learning $\mathcal{C}$ up to $L^2$ error $\epsilon$ (we may pick $\epsilon$ as large as $\beta/3$) requires at least $\mathrm{SDA}_D(\mathcal{C}, \tau^2)$ queries of tolerance $\tau$.*

A recent result of Diakonikolas et al [DKKZ20] gave the following construction of one-layer neural networks on $\mathbb{R}^n$ with $k$ hidden units, i.e. functions of the form $g(x) = \psi(\sum_{i=1}^k a_i \phi(x \cdot w_i))$ for activation functions $\psi, \phi : \mathbb{R} \to \mathbb{R}$ and weights $w_i \in \mathbb{R}^n, a_i \in \mathbb{R}$.

**Theorem 2.3** ([DKKZ20])**.** *There exists a class $\mathcal{G}$ of one-layer neural networks on $\mathbb{R}^n$ with $k$ hidden units such that for some universal constant $0 < c < 1/2$ and $\gamma = n^{\Theta(k(c-1/2))}$, $\mathrm{SDA}(\mathcal{G}, \gamma) \geq 2^{n^c}$.*

*This holds for any $\psi : \mathbb{R} \to [-1, 1]$ that is odd, and $\phi \in L^2(\mathbb{R}, \mathcal{N}(0,1))$ that has a nonzero Hermite coefficient of degree greater than $k/2$. Further, the weights satisfy $|a_i| = 1/k$ and $\|w_i\|_2 = 1$ for all $i$.*

We will be interested in the following special cases. Proofs of the norm lower bounds are in Appendix D.

**Corollary 2.4.** *For the following instantiations of $\mathcal{G}$, with accompanying norm lower bound $\beta$ (i.e. such that $\|g\| \geq \beta$ for all $g \in \mathcal{G}$), there exist $\tau = n^{-\Theta(k)}$ and $\epsilon \geq \tau$ such that learning $\mathcal{G}$ up to $L^2$ error $\epsilon$ requires at least $2^{n^c}$ queries of tolerance $\tau$, for some $0 < c < 1/2$.*

  *(a) ReLU nets: $\psi = \tanh$, $\phi = \mathrm{ReLU}$. Then $\beta = \Omega(1/k^6)$ (Lemma D.4), so we may take $\epsilon = \Theta(1/k^6)$.*
  *(b) Sigmoid nets: $\psi = \tanh$, $\phi = \sigma$. Then $\beta = \exp(-O(\sqrt{k}))$ (Lemma D.6), so we may take $\epsilon = \exp(-\Theta(\sqrt{k}))$.*
  *(c) Majority of halfspaces: $\psi = \phi = \mathrm{sign}$. Being Boolean functions, here $\beta = 1$ exactly, so we may take $\epsilon = \Theta(1)$.*

**Convex Optimization Basics**   Over a general inner product space $\mathcal{Z}$, a function $p : \mathcal{Z} \to \mathbb{R}$ is convex if for all $\alpha \in [0, 1]$ and $z, z' \in \mathcal{Z}$, $p(\alpha z + (1-\alpha)z') \leq \alpha p(z) + (1-\alpha)p(z')$. We say that $s \in \mathcal{Z}$ is a subgradient of $p$ at $z$ if $p(z+h) - p(z) \geq \langle s, h \rangle$. We say that $p$ is $\beta$-smoothly convex if for all $z, h \in \mathcal{Z}$ and any subgradient $s$ of $p$ at $z$,

$$p(z + h) - p(z) - \langle s, h \rangle \leq \frac{\beta}{2}\|h\|^2.$$

If there is a unique subgradient of $p$ at $z$, we simply refer to it as the gradient $\nabla p(z)$. It is easily proven that smoothly convex functions have unique subgradients at all points. Another standard property is the following: for any $z, z' \in \mathcal{Z}$,

$$p(z) - p(z') \leq \langle \nabla p(z), z - z' \rangle - \frac{1}{2\beta}\|\nabla p(z) - \nabla p(z')\|^2. \tag{1}$$

In this paper we will be concerned with convex optimization using the Frank–Wolfe variant of gradient descent, also known as conditional gradient descent. In order to eventually apply this framework to improper learning, we will consider a slight generalization of the standard setup. Let $\mathcal{Z}' \subset \mathcal{Z}$ both be compact, convex subsets of our generic inner product space. Say we have a $\beta$-smoothly convex function $p : \mathcal{Z} \to \mathbb{R}$, and we want to solve $\min_{z \in \mathcal{Z}'} p(z)$, i.e. optimize over the smaller domain, while allowing ourselves the freedom of finding subgradients that lie in the larger $\mathcal{Z}$. The Frank–Wolfe algorithm in this "improper" setting is Algorithm 1.

---

**Algorithm 1** Frank–Wolfe gradient descent over a generic inner product space

---

Start with an arbitrary $z_0 \in \mathcal{Z}$.
**for** $t = 0, \ldots, T$ **do**
    Let $\gamma_t = \frac{2}{t+2}$.
    Find $s \in \mathcal{Z}$ such that $\langle s, -\nabla p(z_t) \rangle \geq \max_{s' \in \mathcal{Z}'} \langle s', -\nabla p(z_t) \rangle - \frac{1}{2}\delta\gamma_t C_p$.
    Let $z_{t+1} = (1 - \gamma_t)z_t + \gamma_t s$.
**end for**

---

The following theorem holds by standard analysis (see e.g. [Jag13]). For convenience, we provide a self-contained proof in Appendix G.

**Theorem 2.5.** *Let $\mathcal{Z}' \subseteq \mathcal{Z}$ be convex sets, and let $p : \mathcal{Z} \to \mathbb{R}$ be a $\beta$-smoothly convex function. Let $C_p = \beta \operatorname{diam}(\mathcal{Z})^2$. For every $t$, the iterates of Algorithm 1 satisfy*

$$p(z_t) - \min_{z' \in \mathcal{Z}'} p(z') \leq \frac{2C_p}{t+2}(1 + \delta).$$

## 3 Functional gradient descent

Let $\ell : \mathbb{R} \times \mathbb{R} \to \mathbb{R}$ be a loss function. Given a $p$-concept $f^*$ and its corresponding labeled distribution $D_{f^*}$, the population loss of a function $f : \mathbb{R}^n \to \mathbb{R}$ is given by $L(f) = \mathbb{E}_{(x,y) \sim D_{f^*}}[\ell(f(x), y)]$. We

will view $L$ as a mapping from $L^2(\mathbb{R}^n, D)$ to $\mathbb{R}$, and refer to it as the loss functional. The general idea of functional gradient descent is to try to find an $f$ in a class of functions $\mathcal{F}$ that minimizes $L(f)$ by performing gradient descent in function space. When using Frank–Wolfe gradient descent, the key step in every iteration is to find the vector that has the greatest projection along the negative gradient, which amounts to solving a linear optimization problem over the domain. When $\mathcal{F}$ is the convex hull $\operatorname{conv}(\mathcal{H})$ of a simpler class $\mathcal{H}$, this can be done using a sufficiently powerful agnostic learning primitive for $\mathcal{H}$. Thus we can "boost" such a primitive in a black-box manner to minimize $L(f)$.

One reason for using Frank–Wolfe rather than say standard projected GD is because in the $L^2$ function space in which we operate, it is not natural to require the learner to find a projection of the functional gradient onto $\operatorname{conv}(\mathcal{H})$. Another important reason is that Frank–Wolfe uses a *linear* optimization subproblem in each step, and this is important in order to carry out the entire reduction purely using statistical queries.

Let $\mathcal{H} \subset L^2(\mathbb{R}^n, D)$ be a base hypothesis class for which we have an agnostic learner with the following guarantee:

**Assumption 3.1.** *There is an SQ learner for $\mathcal{H}$ with the following guarantee. Let $\mathcal{D}$ be any labeled distribution on $\mathbb{R}^n \times \mathbb{R}$ such that the marginal on $\mathbb{R}^n$ is $D = \mathcal{N}(0, I_n)$. Given only SQ access to $\mathcal{D}$, the learner outputs a function $f \in \mathcal{B}(\operatorname{diam}(\mathcal{H})/2)$ such that*

$$\mathbb{E}_{(x,y)\sim\mathcal{D}}[f(x)y] \geq \max_{f'\in\mathcal{H}} \mathbb{E}_{(x,y)\sim\mathcal{D}}[f'(x)y] - \epsilon$$

*using $q(n, \epsilon, \tau)$ queries of tolerance $\tau$.*

*Notice that we do* not *require $f$ to lie in $\mathcal{H}$, i.e. the learner is allowed to be improper, but we do require it to have norm at most $\operatorname{diam}(\mathcal{H})/2$. This is to make the competitive guarantee against $\mathcal{H}$ meaningful, since otherwise the correlation can be made to scale arbitrarily with the norm.*

With such an $\mathcal{H}$ in place, we define $\mathcal{F} = \operatorname{conv}(\mathcal{H})$. We assume that $f^* \in \mathcal{F}$. Our objective will be to agnostically learn $\mathcal{F}$: to solve $\min_{f\in\mathcal{F}} L(f)$ in such a way that $L(\hat{f}) - L(f^*) \leq \epsilon$.

To be able to use Frank–Wolfe, we require some assumptions on the loss function $\ell$.

**Assumption 3.2.** *The loss function $\ell : \mathbb{R} \times \mathbb{R} \to \mathbb{R}$ is $\beta$-smoothly convex in its first parameter.*

As we show in Appendix H, from this assumption it easily follows that the loss functional $L$ is itself $\beta$-smoothly convex, and that the gradient of $L$ at $f$ is given by $\nabla L(f) : x \mapsto \mathbb{E}_{y\sim f^*(x)}[\partial_1 \ell(f(x), y)]$.

**Example 3.3.** The canonical example is the squared loss functional, with $\ell_{\mathsf{sq}}(a, b) = (a - b)^2$, which is 2-smoothly convex. Here the gradient has a very simple form, since $\partial_1 \ell_{\mathsf{sq}}(a, b) = 2(a - b)$, and so

$$\mathbb{E}_{y\sim f^*(x)}[\partial_1 \ell_{\mathsf{sq}}(f(x), y)] = \mathbb{E}_{y\sim f^*(x)}[2(f(x) - y)] = 2(f(x) - f^*(x)),$$

i.e. $\nabla L_{\mathsf{sq}}(f) = 2(f - f^*)$. In fact, it is easily calculated that $L_{\mathsf{sq}}(f) = \mathbb{E}_{(x,y)\sim D_{f^*}}[(f(x) - y)^2] = \|f\|^2 - 2\langle f, f^* \rangle + 1$. It is also useful to note that

$$L_{\mathsf{sq}}(f) - L_{\mathsf{sq}}(f^*) = \|f - f^*\|^2. \tag{2}$$

**Frank–Wolfe using statistical queries**  We see that our loss functional is a $\beta$-smoothly convex functional on the space $L^2(\mathbb{R}^n, D)$. We can now use Frank–Wolfe if we can solve its main subproblem: finding an approximate solution to $\max_{h\in\mathcal{F}}\langle h, -\nabla L(f)\rangle$, where $f$ is the current hypothesis during some iteration. Since this is a linear optimization objective and $\mathcal{F} = \operatorname{conv}(\mathcal{H})$, this is the same as solving $\max_{h\in\mathcal{H}}\langle h, -\nabla L(f)\rangle$. This is almost the guarantee that Assumption 3.1 gives us, but some care is in order. What we have SQ access to is the labeled distribution $D_{f^*}$ on $\mathbb{R}^n \times \{\pm 1\}$. It is not clear that we can rewrite the optimization objective in such a way that

$$\max_{h\in\mathcal{H}} \mathbb{E}_{x\sim D}[-h(x)\nabla L(f)(x)] = \max_{h\in\mathcal{H}} \mathbb{E}_{(x,y')\sim\mathcal{D}}[h(x)y'] \tag{3}$$

for some distribution $\mathcal{D}$ on $\mathbb{R}^n \times \mathbb{R}$ *that we can simulate SQ access to.* Naively, we might try to do this by letting $\mathcal{D}$ be the distribution of $(x, -\nabla L(f)(x))$ for $x \sim D$, so that a query $\phi : \mathbb{R} \times \mathbb{R} \to \mathbb{R}$ to $\mathcal{D}$ can be answered with $\mathbb{E}_{(x,y')\sim\mathcal{D}}[\phi(x, y')] = \mathbb{E}_{x\sim D}[\phi(x, -\nabla L(f)(x))]$. But the issue is that in general $\nabla L(f)(x)$ will depend on $f^*(x)$, which we do not know — all we have access to is $D_{f^*}$.

It turns out that for the loss functions we are interested in, we can indeed find a suitable such $\mathcal{D}$. We turn to the details now.

# 4 Functional gradient descent guarantees on surrogate loss

The functional GD approach applied directly to squared loss would allow us to learn $\mathcal{F} = \text{conv}(\mathcal{H})$ using a learner for $\mathcal{H}$ (that satisfied Assumption 3.1). But by considering a certain surrogate loss, we can use the same learner to actually learn $\psi \circ \mathcal{F} = \{\psi \circ f \mid f \in \mathcal{F}\}$ for an outer activation function $\psi$. This is particularly useful as we can now capture $p$-concepts corresponding to functions in $\mathcal{F}$ by using a suitable $\psi : \mathbb{R} \to [-1, 1]$. For example, the common softmax activation corresponds to taking $\psi = \tanh$.

Assume that $\mathbb{E}[y|x] = \psi(f^*(x))$ for some activation $\psi : \mathbb{R} \to \mathbb{R}$ which is non-decreasing and $\lambda$-Lipschitz. Instead of the squared loss, we will consider the following surrogate loss:

$$\ell_{\mathsf{sur}}(a, b) = \int_0^a (\psi(u) - b) du.$$

One can show that $\ell_{\mathsf{sur}}$ is in fact $\lambda$-smoothly convex, and that $\partial_1 \ell_{\mathsf{sur}}(a, b) = \psi(a) - b$. The gradient of the surrogate loss functional, $L_{\mathsf{sur}}(f) = \mathbb{E}_{(x,y) \sim D_{\psi \circ f^*}}[\ell_{\mathsf{sur}}(f(x), y)]$, is given by

$$\nabla L_{\mathsf{sur}}(f) : x \mapsto \mathbb{E}_{y \sim \psi(f^*(x))}[\partial_1 \ell_{\mathsf{sur}}(f(x), y)] = \psi(f(x)) - \psi(f^*(x)),$$

i.e. $\nabla L_{\mathsf{sur}}(f) = \psi \circ f - \psi \circ f^*$.

We still need to show that the Frank–Wolfe subproblem can be solved using access to just $D_{\psi \circ f^*}$. Observe that

$$
\begin{aligned}
\mathbb{E}_{x \sim D}[-h(x)\nabla L_{\mathsf{sur}}(f)(x)] &= \mathbb{E}_{x \sim D}[h(x)(\psi(f^*(x)) - \psi(f(x)))] \\
&= \mathbb{E}_{x \sim D}\left[ h(x) \left( \mathbb{E}_{y \sim \psi(f^*(x))}[y] - \psi(f(x)) \right) \right] \\
&= \mathbb{E}_{(x,y) \sim D_{\psi \circ f^*}}[h(x)(y - \psi(f(x)))] \\
&= \mathbb{E}_{(x,y') \sim \mathcal{D}}[h(x)y'],
\end{aligned}
$$

where $\mathcal{D}$ is the distribution of $(x, y - \psi(f(x)))$ for $(x, y) \sim D_{\psi \circ f^*}$. We can easily simulate SQ access to this using $D_{\psi \circ f^*}$: if $\phi$ is any query to $\mathcal{D}$, then

$$\mathbb{E}_{(x,y') \sim \mathcal{D}}[\phi(x, y')] = \mathbb{E}_{(x,y) \sim D_{\psi \circ f^*}}[\phi(x, y - \psi(f(x)))] = \mathbb{E}_{(x,y) \sim D_{\psi \circ f^*}}[\phi'(x, y)] \tag{4}$$

for the modified query $\phi'(x, y) = \phi(x, y - \psi(f(x)))$. This means we can rewrite the optimization objective to fit the form in Eq. (3). Thus for our surrogate loss, Assumption 3.1 allows us to solve the Frank–Wolfe subproblem, giving us Algorithm 2 for learning $\mathcal{F}$.

---

**Algorithm 2** Frank–Wolfe for solving $\min_{f \in \mathcal{F}} L_{\mathsf{sur}}(f)$

---

Start with an arbitrary $f_0 \in \mathcal{B}(\text{diam}(\mathcal{H})/2)$.
**for** $t = 0, \ldots, T$ **do**
  Let $\gamma_t$ be $\frac{2}{t+2}$.
  Let $\mathcal{D}_t$ be the distribution of $(x, y - \psi(f_t(x)))$ for $(x, y) \sim D_{\psi \circ f^*}$.
  Using Assumption 3.1, find $h \in \mathcal{B}(\text{diam}(\mathcal{H})/2)$ such that

$$\mathbb{E}_{(x,y') \sim \mathcal{D}_t}[h(x)y'] \geq \max_{h' \in \mathcal{H}} \mathbb{E}_{(x,y') \sim \mathcal{D}_t}[h'(x)y'] - \frac{1}{2}\gamma_t \lambda \, \text{diam}(\mathcal{H})^2$$

  Let $f_{t+1} = (1 - \gamma_t)f_t + \gamma_t h$.
**end for**

---

**Theorem 4.1.** *Let $\mathcal{H}$ be a class for which Assumption 3.1 holds, and let $\mathcal{F} = \text{conv}(\mathcal{H})$. Given SQ access to $D_{\psi \circ f^*}$ for a known non-decreasing $\lambda$-Lipschitz activation $\psi$ and an unknown $f^* \in \mathcal{F}$, suppose we wish to learn $\psi \circ f^*$ in terms of surrogate loss, i.e. to minimize $L_{\mathsf{sur}}(f)$. Then after $T$ iterations of Algorithm 2, we have the following guarantee:*

$$L_{\mathsf{sur}}(f_T) - L_{\mathsf{sur}}(f^*) \leq \frac{4\lambda \, \text{diam}(\mathcal{H})^2}{T + 2}.$$

*In particular, we can achieve $L_{\text{sur}}(f_T) - L_{\text{sur}}(f^*) \leq \epsilon$ after $T = O(\frac{\lambda \operatorname{diam}(\mathcal{H})^2}{\epsilon})$ iterations. Assuming our queries are of tolerance $\tau$, the total number of queries used is at most $Tq(n, \frac{\epsilon}{4}, \tau) = O(\frac{\lambda \operatorname{diam}(\mathcal{H})^2}{\epsilon} q(n, \frac{\epsilon}{4}, \tau))$.*

*Proof.* By the preceding discussion, the surrogate loss functional is $\lambda$-smoothly convex, and Algorithm 2 is a valid special case of Algorithm 1, with $\mathcal{Z} = \mathcal{B}(\operatorname{diam}(\mathcal{H})/2)$ and $\mathcal{Z}' = \operatorname{conv}(\mathcal{F})$. Thus the guarantee follows directly from Theorem 2.5 (setting $\delta = 1$).

To bound the number of queries, observe that it is sufficient to run for $T = \frac{4\lambda \operatorname{diam}(\mathcal{H})^2}{\epsilon} - 2$ rounds. In the $t^{\text{th}}$ iteration, we invoke Assumption 3.1 with

$$\epsilon' = \frac{1}{2}\gamma_t \lambda \operatorname{diam}(\mathcal{H})^2 = \frac{\lambda \operatorname{diam}(\mathcal{H})^2}{t+2} \geq \frac{\lambda \operatorname{diam}(\mathcal{H})^2}{T+2} = \frac{\epsilon}{4}.$$

Since $q(n, \epsilon', \tau) \leq q(n, \frac{\epsilon}{4}, \tau)$, the bound follows. $\qquad\square$

Lastly, we can show that minimizing surrogate loss also minimizes the squared loss. Observe first that $\nabla L_{\text{sur}}(f^*) = 0$. Thus, applying Eq. (1) with $z = f^*$ and $z' = f$, we get

$$
\begin{aligned}
L_{\text{sur}}(f) - L_{\text{sur}}(f^*) &\geq \frac{1}{2\lambda} \|\nabla L_{\text{sur}}(f) - \nabla L_{\text{sur}}(f^*)\|^2 \\
&= \frac{1}{2\lambda} \|\psi \circ f - \psi \circ f^*\|^2 \\
&= \frac{1}{2\lambda}(L_{\text{sq}}(\psi \circ f) - L_{\text{sq}}(\psi \circ f^*)),
\end{aligned}
\tag{5}
$$

where $L_{\text{sq}}$ is squared loss wrt $D_{\psi \circ f^*}$ and the last equality is Eq. (2). In particular, Eq. (5) implies that $\psi \circ f$ achieves the following $L^2$ error with respect to $\psi \circ f^*$:

$$\|\psi \circ f - \psi \circ f^*\| \leq \sqrt{2\lambda(L_{\text{sur}}(f) - L_{\text{sur}}(f^*))}. \tag{6}$$

# 5    Lower bounds on learning ReLUs, sigmoids, and halfspaces

The machinery so far has shown that if we could agnostically learn a single unit (e.g. a ReLU or a sigmoid), we could learn depth-two neural networks composed of such units. Since we have lower bounds on the latter problem, this yields the following lower bounds on the former.

**Theorem 5.1.** *Let $\mathcal{H}_{\text{ReLU}} = \{x \mapsto \pm \operatorname{ReLU}(w \cdot x) \mid \|w\|_2 \leq 1\}$ be the class of ReLUs on $\mathbb{R}^n$ with unit weight vectors.[1] Suppose that Assumption 3.1 holds for $\mathcal{H}_{\text{ReLU}}$. Then for any $\epsilon$, there exists $\tau = n^{-\Theta(\epsilon^{-1/12})}$ such that $q(n, \epsilon, \tau) \geq 2^{n^c}\epsilon$ for some $0 < c < 1/2$.*

*Proof.* Since all our lower bound proofs are similar, to set a template we lay out all the steps as clearly as possible.

- Consider the class $\mathcal{G}$ from Theorem 2.3 instantiated with $\psi = \tanh$ (which is 1-Lipschitz, so $\lambda = 1$) and $\phi = \operatorname{ReLU}$. By the conditions on the weights, we see that $\mathcal{G} \subseteq \tanh \circ \mathcal{F}_{\text{ReLU}}$, where $\mathcal{F}_{\text{ReLU}} = \operatorname{conv}(\mathcal{H}_{\text{ReLU}})$. This construction has a free parameter $k$, which we will set based on $\epsilon$.
- By our main reduction (Assumption 3.1 and Theorem 4.1), we can learn $\tanh \circ \mathcal{F}_{\text{ReLU}}$ with respect to $L_{\text{sur}}$ up to agnostic error $\epsilon$ using $O(\frac{1}{\epsilon} q(n, \frac{\epsilon}{4}, \tau))$ queries of tolerance $\tau$. By Eq. (6), this implies learning $\mathcal{G}$ up to $L^2$ error $\sqrt{2\epsilon}$.
- We know that learning $\mathcal{G}$ should be hard. Specifically, Corollary 2.4(a) states that if $\epsilon' = \Theta(1/k^6)$ and the queries are of tolerance $\tau = n^{-\Theta(k)}$, then learning up to $L^2$ error $\epsilon'$ should require $2^{n^c}$ queries.
- The loss our reduction achieves is $\epsilon' = \sqrt{2\epsilon}$, so we require $\sqrt{2\epsilon} \leq \Theta(1/k^6)$ for the bound to hold. Accordingly, we pick $k = \Theta(\epsilon^{-1/12})$, so that $\tau = n^{-\Theta(k)} = n^{-\Theta(\epsilon^{-1/12})}$.
- Thus we must have $\frac{1}{\epsilon} q(n, \frac{\epsilon}{4}, \tau) \geq 2^{n^c}$. Rearranging and rescaling $\epsilon$ gives the result.

$\square$

**Theorem 5.2.** *Let* $\mathcal{H}_\sigma = \{x \mapsto \pm\sigma(w \cdot x) \mid \|w\|_2 \leq 1\}$, *where* $\sigma$ *is the standard sigmoid, be the class of sigmoid units on* $\mathbb{R}^n$ *with unit weight vectors. Suppose that Assumption 3.1 holds for* $\mathcal{H}_\sigma$. *Then for any* $\epsilon$, *there exists* $\tau = n^{-\Theta((\log 1/\epsilon)^2)}$ *such that* $q(n, \epsilon, \tau) \geq 2^{n^c}\epsilon$ *for some* $0 < c < 1/2$.

*Proof.* Very similar to the above. We instantiate $\mathcal{G}$ with $\psi = \tanh$, $\phi = \sigma$, and observe that $\mathcal{G} \subseteq \tanh \circ \operatorname{conv}(\mathcal{H}_\sigma)$ and that $\operatorname{diam}(\mathcal{H}_\sigma) \leq 2$. In this case, Corollary 2.4(b) tells us that we require $\sqrt{2\epsilon} \leq e^{-\Theta(\sqrt{k})}$ for the lower bound to hold, so we pick $k = (\log 1/\epsilon)^2$. The result now follows exactly as before. $\square$

We also obtain a lower bound on the class of halfspaces. Note that for Boolean functions, functional GD is not essential; existing distribution-specific boosting methods [KK09, Fel10] can also give us similar results here.

**Theorem 5.3.** *Let* $\mathcal{H}_{\mathsf{hs}} = \{x \mapsto \operatorname{sign}(w \cdot x) \mid \|w\|_2 \leq 1\}$ *be the class of halfspaces on* $\mathbb{R}^n$ *with unit weight vectors. Suppose that Assumption 3.1 holds for* $\mathcal{H}_{\mathsf{hs}}$. *Then for any* $\epsilon$, *there exists* $\tau = n^{-\Theta(1/\epsilon)}$ *such that* $q(n, \epsilon, \tau) \geq 2^{n^c}\epsilon^3$ *for some* $0 < c < 1/2$.

*Proof.* To approximate the sign function using a Lipschitz function, we define $\widetilde{\operatorname{sign}}(x)$ to be $-1$ for $x \leq -1/k$, $1$ for $x \geq 1/k$, and linearly interpolate in between. This function is $(k/2)$-Lipschitz. We claim that $\mathcal{G}$ instantiated with $\psi = \phi = \operatorname{sign}$ satisfies $\mathcal{G} \subseteq \widetilde{\operatorname{sign}} \circ \operatorname{conv}(\mathcal{H}_{\mathsf{hs}})$, with $\operatorname{diam}(\mathcal{G}) = 2$. This is because as noted in Theorem 2.3, $\mathcal{G}$ has weights $a_i \in \{\pm 1/k\}$, so the sum of halfspaces inside $\psi$ is always a multiple of $1/k$, and $\widetilde{\operatorname{sign}}$ behaves the same as $\operatorname{sign}$.

Theorem 4.1 now lets us learn $\mathcal{G}$ up to agnostic error $\epsilon$ (and hence $L^2$ error $\sqrt{2k\epsilon}$, by $Eq.$ (6)) using $O(\frac{k^2}{\epsilon}q(n, \frac{\epsilon}{4}, \tau))$ queries of tolerance $\tau$. By Corollary 2.4(c), we only need $\sqrt{2k\epsilon} \leq \Theta(1)$ for the lower bound to hold, so we may take $k = \Theta(1/\epsilon)$ to get a lower bound of $2^{n^c}$. Thus $\frac{k^2}{\epsilon}q(n, \frac{\epsilon}{4}, \tau) \geq 2^{n^c}$, and rearrangement gives the result. $\square$

## 6 Upper bounds on learning ReLUs and sigmoids

We use a variant of the classic low-degree algorithm ([LMN93]; see also [KKMS08]) to provide simple upper bounds for agnostically learning ReLUs and sigmoids. With respect to $D = \mathcal{N}(0, I_n)$, the $\delta$-approximate degree of a function $f : \mathbb{R}^n \to \mathbb{R}$ is the smallest $d$ such that there exists a degree-$d$ polynomial $p$ satisfying $\|f - p\| \leq \delta$. In Appendix E, we show that for any class $\mathcal{C}$ of $\delta$-approximate degree $d$, picking $\delta = O(\epsilon)$ and simply estimating the Hermite coefficients of $x \mapsto \mathbb{E}[y|x]$ up to degree $d$ yields an agnostic learner up to error $\epsilon$. For simplicity we assume that $y \in [-1, 1]$.

**Theorem 6.1.** *The class* $\mathcal{H}_{\mathrm{ReLU}}$ *can be agnostically learned up to* $\epsilon$ *using* $n^{O(\epsilon^{-4/3})}$ *queries of tolerance* $n^{-\Theta(\epsilon^{-4/3})}\epsilon$. *Similarly,* $\mathcal{H}_\sigma$ *can be learned using* $n^{\tilde{O}(\log^2 1/\epsilon)}$ *queries of tolerance* $n^{-\tilde{\Theta}(\log^2 1/\epsilon)}\epsilon^2$.

*Proof.* Approximating the Hermite coefficients of degree at most $d$ takes $n^{O(d)}$ queries of tolerance $n^{-\Theta(d)}\epsilon$. As we show in Appendix F, the $\delta$-approximate degree of unit-weight ReLUs is $O((1/\delta)^{4/3})$ of unit-weight sigmoids is $\tilde{O}(\log^2 1/\delta)$. The guarantees follow by the argument in Appendix E. $\square$

We note that our lower bounds for ReLUs and sigmoids were for queries of tolerance $n^{-\Theta(\epsilon^{-1/12})}$ and $n^{-\Theta(\log^2 1/\epsilon)}$ respectively, which nearly matches these upper bounds.

## Acknowledgements

We thank the anonymous reviewers for their feedback.

## Broader Impact

As deep learning techniques continue to grow rapidly in real-world usage and importance, the study of their theoretical guarantees has become important as well. This paper gives theoretical results concerning some basic primitives of neural networks, and as such contributes to our growing rigorous understanding of deep learning. The lower bounds in this paper address a model of learning in which there is arbitrary noise in the labels, as is common in some settings. They add to the body of work demonstrating that even very simple neural networks can fail dramatically to have good theoretical guarantees, providing some guidance as to conditions under which neural networks might perform particularly poorly even in practice.

Another area of practical importance that our results relate to is that of differentially private data analysis/release, where the SQ complexity of agnostic learning is known to characterize the complexity of privately answering a class of queries up to small error [GHRU13, DR14].

## Funding

SG was supported by the JP Morgan AI PhD Fellowship. AG was supported by NSF awards AF-1909204 and AF-1717896, and a UT Austin Provost's Fellowship. AK was supported by NSF awards AF-1909204, AF-1717896, and the NSF AI Institute for Foundations of Machine Learning (IFML). Work done while visiting the Institute for Advanced Study, Princeton, NJ.

## Footnotes

[1] We use $\pm\operatorname{ReLU}$ for simplicity. Any learner can handle this by doing a bit flip on its own.

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
