[Supplementary Material]

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

_{\mathsf{sur}}(f) - L_{\mathsf{sur}}(f^*) &\geq \frac{1}{2\lambda} \|\nabla L_{\mathsf{sur}}(f) - \nabla L_{\mathsf{sur}}(f^*)\|^2 \\
&= \frac{1}{2\lambda} \|\psi \circ f - \psi \circ f^*\|^2 \\
&= \frac{1}{2\lambda}(L_{\mathsf{sq}}(\psi \circ f) - L_{\mathsf{sq}}(\psi \circ f^*)),
\end{aligned}
\tag{5}
$$

where $L_{\mathsf{sq}}$ is squared loss wrt $D_{\psi \circ f^*}$ and the last equality is Eq. (2). In particular, Eq. (5) implies that $\psi \circ f$ achieves the following $L^2$ error with respect to $\psi \circ f^*$:

$$\|\psi \circ f - \psi \circ f^*\| \leq \sqrt{2\lambda\left(L_{\mathsf{sur}}(f) - L_{\mathsf{sur}}(f^*)\right)}. \tag{6}$$

## 5 Lower bounds on learning ReLUs, sigmoids, and halfspaces

The machinery so far has shown that if we could agnostically learn a single unit (e.g. a ReLU or a sigmoid), we could learn depth-two neural networks composed of such units. Since we have lower bounds on the latter problem, this yields the following lower bounds on the former.

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

[2]Note that here we are assuming $\langle h_{\mathsf{sq}}, f_{\mathsf{cmf}}\rangle \geq 0$ WLOG, since otherwise we would consider $-h_{\mathsf{sq}}$.

[3]For another way to see this, for any nonzero $h \in \mathcal{H}$, expand $\|\lambda h - f_{\mathsf{cmf}}\|^2 \geq \|0 - f_{\mathsf{cmf}}\|^2$ and let $\lambda \to 0$.

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

# A SQ lower bound subtleties

## A.1 Relationships between parameters

When formally stating SQ lower bounds on learning $p$-concepts in terms of the statistical dimension, there are some subtleties to keep in mind. These have to do with the relationships between the query tolerance, the desired final error, and the norms of the functions in the class. Let us say our queries are of tolerance $\tau$, the final desired $L^2$ error $\|f - f^*\|$ is $\epsilon$ (which corresponds to $L(f) - L(f^*) \leq \epsilon^2$; see Eq. (2)), and that the functions in $\mathcal{C}$ satisfy $\|f^*\| \geq \beta$ for all $f^* \in \mathcal{C}$. Then

1. We must have $\tau < \epsilon$. To see why, first note that for any query $\phi$ and two functions $f, g \in \mathcal{C}$, a calculation shows that $|\mathbb{E}_{D_f}[\phi] - \mathbb{E}_{D_g}[\phi]| = |\langle f - g, \tilde{\phi} \rangle| \leq \|f - g\|$, where $\tilde{\phi}(x) = (\phi(x, 1) - \phi(x, -1))/2$. Thus if one has a function $f$ such that $\epsilon < \|f - f^*\| < \tau$, then no query of tolerance $\tau$ can tell them apart, but $f$ is not $\epsilon$-close to the target $f^*$.

2. If $\epsilon \geq \beta$, a lower bound might not be possible. This is because the 0 function trivially achieves $L^2$ error $\|0 - f^*\| = \|f^*\|$. Imposing $\epsilon < \beta$ is sufficient to rule this out.

3. We cannot arbitrarily rescale the p-concepts to increase $\beta$ since the functions must remain Boolean $p$-concepts. Rescaling would also increase the description length of the functions.

The lower bound in Theorem 2.2 (from [GGJ$^+$20]) is proved by reducing a distinguishing problem to a learning problem. For technical reasons, we end up requiring $\tau \leq \epsilon^2$, $\epsilon \leq \beta/3$ for this reduction to go through. The points above show that these requirements are essentially necessary.

## A.2 The dependence of the query lower bound on the error $\epsilon$ and the tolerance $\tau$

The relationship between our query lower bounds, the desired error $\epsilon$, and the tolerance $\tau$ may seem a little unusual at first sight, especially the fact that the lower bounds seem to grow weaker as $\epsilon$ grows smaller. We make some clarifying remarks here.

Fundamentally, all SQ lower bounds are bounds on how many queries it takes to distinguish certain distributions from others. When discussing a concept class $\mathcal{C}$, the distributions in question are the labeled distributions corresponding to concepts in the class. Learning $\mathcal{C}$ is hard exactly insofar as it allows us to distinguish different labeled distributions arising from $\mathcal{C}$. Many works in the SQ literature have this structure, but we will refer to [GGJ$^+$20] for formal statements.

Formally, the distinguishing problem we consider ([GGJ$^+$20, Definition 4.2]) is that of distinguishing the labeled distribution $D_c$ arising from an unknown $c \in \mathcal{C}$ from the reference distribution $D_0 = D \times \text{Unif}\{\pm 1\}$, using queries of tolerance at least $\tau$.

There are two crucial points to keep in mind here:

1. The distinguishing problem is a fundamentally information theoretic problem, and its difficulty scales only with $\tau$. In particular, using queries of tolerance $\tau$, we need at least $\text{SDA}(\mathcal{C}, \tau^2)$ queries. This bound increases with $\tau$; in fact it often scales as $|\mathcal{C}|\tau^2$ (see ([GGJ$^+$20, Theorem 4.5 and Lemma 2.6]).

2. The problem of learning up $\mathcal{C}$ to error $\epsilon$ is hard exactly insofar as it allows us to solve the distinguishing problem (see [GGJ$^+$20, Lemma 4.4]).

An important consequence is that for fixed $\tau$, the query lower bound does not technically grow as a function of the error $\epsilon$: it applies uniformly for all $\epsilon$ small enough that it allows the learner to solve the distinguishing problem. In other words, there is a certain "threshold" $\epsilon_0$ such that for all $\epsilon \leq \epsilon_0$, the same query lower bound holds. As noted in point (3) of the previous subsection, this threshold can be taken to be $\beta/3$, where $\beta$ is such that $\|c\| \geq \beta$ for all $c \in \mathcal{C}$.

But at the same time, as noted in point (1) in the previous subsection, it is necessary that $\tau < \epsilon$ (and for the reduction it suffices to have $\tau \leq \epsilon^2$). If $\tau \geq \epsilon$, learning up to error $\epsilon$ is simply impossible.

With all this in mind, we can now answer the question of why our lower bounds seem to grow weaker as $\epsilon$ grows smaller: it is essentially because $\tau$ grows smaller as well, so that we get a series of incomparable (though still exponential) bounds due to the tradeoffs between query complexity, $\tau$, and $\epsilon$.

# B  Lower bounds on learning monomials

Let $\mathcal{H}_{\mathsf{mon}}$ be the class of all multilinear monomials of total degree $d$ on $\mathbb{R}^n$. Clearly $|\mathcal{H}_{\mathsf{mon}}| = \binom{n}{d} = n^{\Theta(d)}$. For any two distinct multilinear monomials $f, g$, clearly $\langle f, g \rangle = 0$ and moreover $\langle \tanh \circ f, \tanh \circ g \rangle = 0$ as well. Thus the class $\mathcal{G} = \tanh \circ \mathcal{H}_{\mathsf{mon}}$ consists entirely of orthogonal functions. By [GGJ$^+$20, Lemma 2.6], $\mathrm{SDA}(\mathcal{G}, \gamma) \geq |\mathcal{G}|\gamma = n^{-\Theta(d)}\gamma$.

We still need a norm lower bound on $\mathcal{G}$.

**Lemma B.1.** *Let $x_S = \prod_{i \in S} x_i$ be an arbitrary degree-$d$ multilinear monomial on $\mathbb{R}^n$, where $S \subseteq [n]$ is a subset of size $d$. Then $\|\tanh \circ x_S\| \geq \exp(-\Theta(d))$.*

*Proof.* Observe first that $\|x_S\| = 1$. By Paley–Zygmund, we have

$$\mathbb{P}[x_S^2 \geq \theta \, \mathbb{E}[x_S^2]] \geq (1 - \theta)^2 \frac{\mathbb{E}[x_S^2]^2}{\mathbb{E}[x_S^4]}.$$

By picking $\theta = 1/2$, say, and using the fact that by Gaussian hypercontractivity,

$$\frac{\mathbb{E}[x_S^2]^2}{\mathbb{E}[x_S^4]} = \prod_{i \in S} \frac{\mathbb{E}[x_i^2]^2}{\mathbb{E}[x_i^4]} \geq \exp(-\Theta(d)),$$

we get that $\mathbb{P}[|x_S| \geq 1/2] \geq \exp(-\Theta(d))$.

Now since $\tanh$ is monotonic and odd, we have

$$\mathbb{E}[\tanh(x_S)^2] \geq \tanh(1/2)^2 \, \mathbb{P}[|x_S| \geq 1/2] \geq \exp(-\Theta(d)).$$

$\square$

By Theorem 2.2 with $\beta = \exp(-\Theta(d))$, we get that for any $\epsilon \leq \exp(-\Theta(d))$ and using queries of tolerance $\tau \leq \epsilon^2$, learning $\mathcal{G}$ up to $L^2$ error $\epsilon$ takes at least $\mathrm{SDA}(\mathcal{G}, \tau^2) \geq n^{\Theta(d)}\tau^2$ queries.

Now we can use the same arguments as in Section 5 to prove the following.

**Theorem B.2.** *Suppose that Assumption 3.1 holds for $\mathcal{H}_{\mathsf{mon}}$. Then for any $\epsilon \leq \exp(-\Theta(d))$ and $\tau \leq \epsilon^2$, $q(n, \epsilon, \tau) \geq n^{\Theta(d)}\tau^{5/2}$.*

*Proof.* Observe that $\mathcal{G} \subseteq \tanh \circ \mathrm{conv}(\mathcal{H}_{\mathsf{mon}})$, and $\mathrm{diam}(\mathcal{H}_{\mathsf{mon}}) \leq 2$. Using the surrogate loss with $\psi = \tanh$, Assumption 3.1 and Theorem 4.1 tell us that we can learn $\tanh \circ \mathrm{conv}(\mathcal{H}_{\mathsf{mon}})$ up to $L^2$ error $\sqrt{2\epsilon}$ (again by Eq. (2)) in $O(\frac{1}{\epsilon}q(n, \epsilon, \tau))$ queries of tolerance $\tau$. By our lower bound for $\mathcal{G}$, we must have $\frac{1}{\epsilon}q(n, \epsilon, \tau) \geq n^{\Theta(d)}\tau^2$, or $q(n, \epsilon, \tau) \geq n^{\Theta(d)}\tau^{5/2}$ (since $\epsilon \geq \sqrt{\tau}$).  $\square$

# C  Lower bounds on learning general non-polynomial activations

Here we extend our lower bounds to general non-polynomial activations $\phi : \mathbb{R} \to \mathbb{R}$, by which we mean functions which have an infinite Hermite series $\phi = \sum_a \widehat{\phi}_a H_a$, where the $H_a$ are the normalized probabilists' Hermite polynomials. We will again work with the [DKKZ20] construction $\mathcal{G}$, instantiated with this $\phi$ and $\psi = \tanh$. In Appendix D, we define this construction formally, letting $g$ be the inner function and $f$ be $\psi \circ g$.

To apply our framework, we need a norm lower bound on $f$. In Lemma D.1 we show that $\|g\|$ is determined only by $k$, the number of hidden units (there $k = 2m$), and the Hermite expansion of $\phi$. The reason we require an infinite Hermite series for $\phi$ is so that this lower bound, viewed as a function of $k$, is nonzero for infinitely many $k$. This then implies that $f = \tanh \circ g$ must be nonzero for infinitely many $k$. Its norm can only possibly be a function of $\phi$ and $k$. In particular, we may assume that it satisfies a norm lower bound $\|f\| \geq \beta(k)$, where $\beta$ is a function only of $k$ that is nonzero for infinitely many $k$. Here we view the dependence on $\phi$ as constant.

A few remarks are in order as to how such a bound $\beta(k)$ may be quantitatively established. If $\phi$ is either bounded or exhibits only polynomial growth, then the bound on $\|g\|$ (Lemma D.1) gives a corresponding lower bound on $\|f\|$ that is also purely a function of $k$. If $\phi$ is bounded, the calculation

is straightforward and very similar to the $\phi = \sigma$ case (Lemma D.6). If $\phi$ grows only like a polynomial, then one can use a truncation argument similar to the $\phi = \text{ReLU}$ case (Lemma D.4).

By Theorem 2.2 and Corollary 2.4, our lower bound of $2^{n^c}$ on learning $\mathcal{G}$ holds for $\epsilon \le \beta(k)/3$. Since we can pick $k$ as we like, let us say that for all sufficiently small $\epsilon$, we can achieve $\epsilon \le \beta(k)/3$ by taking $k = k(\epsilon) = 3\beta^{-1}(\epsilon)$. The corresponding tolerance is then $\tau = n^{-\Theta(k(\epsilon))}$, which is still inverse superpolynomial in $n$.

We now get the following lower bound on learning $\mathcal{H} = \{x \mapsto \phi(w \cdot x) \mid \|w\|_2 \le 1\}$, again by the same arguments as in Section 5. We assume that $\|\phi\| \le R$ for some $R$, so that $\text{diam}(\mathcal{H}) \le 2R$.

**Theorem C.1.** *Suppose that Assumption 3.1 holds for $\mathcal{H}$. Then for all sufficiently small $\epsilon$ and $\tau = n^{-\Theta(k(\epsilon))}$, $q(n, \epsilon, \tau) \ge 2^{n^c} \frac{\epsilon}{R^2}$ for some $0 < c < 1/2$.*

*Proof.* We have $\mathcal{G} \subseteq \tanh \circ \text{conv}(\mathcal{H})$. By functional GD wrt surrogate loss (Theorem 4.1), we see that we can learn $\mathcal{G}$ up to $L^2$ error $\sqrt{2\epsilon}$ using $O(\frac{R^2}{\epsilon} q(n, \epsilon, \tau))$ queries of tolerance $\tau$, but we must have $O(\frac{R^2}{\epsilon} q(n, \epsilon, \tau)) \le 2^{n^c}$. $\qquad\square$

# D    Bounding the function norms of the [DKKZ20] construction

We shall consider the following slight rescaling of the functions of [DKKZ20]. For activation functions $\psi, \phi : \mathbb{R} \to \mathbb{R}$, we have $g, f : \mathbb{R}^2 \to \mathbb{R}$ defined as follows.

$$g(x) = \frac{1}{2m} \sum_{i=1}^{2m} (-1)^i \phi\left(x_1 \cos\frac{i\pi}{m} + x_2 \sin\frac{i\pi}{m}\right) = \frac{1}{2m} \sum_{i=1}^{2m} (-1)^i \phi(x \cdot w_i)$$

$$f(x) = \psi(g(x)),$$

where $w_i = (\cos\frac{i\pi}{m}, \sin\frac{i\pi}{m})$. The number of hidden units is $k = 2m$. We will assume that $m$ is even.

The hard functions from $\mathbb{R}^n \to \mathbb{R}$ are then given by $f_A(x) = f(Ax)$ for certain matrices $A \in \mathbb{R}^{2 \times d}$ with $AA^T = I_2$. For $x \sim \mathcal{N}(0, I_d)$, $Ax$ has the distribution $\mathcal{N}(0, I_2)$. So for the purposes of the norm calculation, and hence throughout this section, we will work directly with $\mathcal{N}(0, I_2)$. We will start by considering the norm of $g$. This can then be used to control the norm of $f$ via arguments similar to those in [GGJ+20].

**Lemma D.1.** *Let $g : \mathbb{R}^2 \to \mathbb{R}$ be as defined above, and assume $m$ is even. Assume the standard Hermite expansion of $\phi$ is given by $\phi = \sum_a \widehat{\phi}_a H_a$, where the $H_a$ are the normalized probabilists' Hermite polynomials. Under $\mathcal{N}(0, I_2)$,*

$$\|g\|^2 = \Omega\left(\sum_{\substack{a \gg m \\ a \text{ even}}} \frac{\widehat{\phi}_a^2}{\sqrt{a}}\right).$$

*(For practical purposes, the asymptotic behavior of this expression is captured faithfully when we begin indexing from say $a = 100m$.)*

*Proof.* We have

$$\|g\|^2 = \mathbb{E}[g(x)^2] = \frac{1}{4m^2} \sum_{i,j=1}^{2m} (-1)^i (-1)^j \mathbb{E}[\phi(x \cdot w_i)\phi(x \cdot w_j)]$$

$$= \frac{1}{4m^2} \sum_{i,j=1}^{2m} (-1)^i (-1)^j \mathbb{E}\left[\left(\sum_a \widehat{\phi}_a H_a(x \cdot w_i)\right)\left(\sum_b \phi_b H_b(x \cdot w_j)\right)\right]$$

$$= \frac{1}{4m^2} \sum_{i,j=1}^{2m} (-1)^i (-1)^j \left(\sum_a \widehat{\phi}_a^2 \mathbb{E}[H_a(x \cdot w_i)H_a(x \cdot w_j)]\right).$$

Now because $w_i, w_j$ are both unit vectors with $w_i \cdot w_j = \cos \frac{(i-j)\pi}{m}$, we have that $x \cdot w_i$ and $x \cdot w_j$ are both $\mathcal{N}(0, 1)$ with covariance $\cos \frac{(i-j)\pi}{m}$. Thus

$$\|g\|^2 = \frac{1}{4m^2} \sum_{i,j=1}^{2m} (-1)^{i+j} \left( \sum_a \widehat{\phi}_a^2 \cos^a \frac{(i-j)\pi}{m} \right)$$

$$= \frac{1}{4m^2} \sum_{i,j=1}^{2m} (-1)^{i-j} \left( \sum_a \widehat{\phi}_a^2 \cos^a \frac{(i-j)\pi}{m} \right),$$

since $(-1)^{i+j} = (-1)^{i-j}$. Now, as we range over $i, j \in [2m]$, we see that $i - j = 0$ occurs $2m$ times, $i - j = 1$ occurs $2m - 1$ times, and more generally $i - j = t$ occurs $2m - |t|$ times. Since a term with $i - j = t$ is exactly the same as one with $i - j = -t$ (by the evenness of $\cos$), we can say that for $t \neq 0$, $|i - j| = t$ occurs $2(2m - t)$ times. Thus the expression above can be written as

$$\|g\|^2 = \frac{1}{4m^2} \left( 2m \left( \sum_a \widehat{\phi}_a^2 \cos^a 0 \right) + \sum_{t=1}^{2m-1} 2(2m - t)(-1)^t \left( \sum_a \widehat{\phi}_a^2 \cos^a \frac{t\pi}{m} \right) \right)$$

$$= \frac{1}{4m^2} \sum_a \widehat{\phi}_a^2 \left( 2m + \sum_{t=1}^{2m-1} 2(2m - t)(-1)^t \cos^a \frac{t\pi}{m} \right)$$

$$= \frac{1}{4m^2} \sum_a \widehat{\phi}_a^2 S(a, m), \tag{7}$$

where

$$S(a, m) = 2m + \sum_{t=1}^{2m-1} 2(2m - t)(-1)^t \cos^a \frac{t\pi}{m}.$$

Now some algebraic manipulations are in order. By rewriting the index $t$ as $2m - t$, we get that

$$S(a, m) = 2m + \sum_{t=1}^{2m-1} 2t(-1)^{2m-t} \cos^a \frac{(2m - t)\pi}{m}$$

$$= 2m + \sum_{t=1}^{2m-1} 2t(-1)^t \cos^a \frac{t\pi}{m}.$$

Adding the two expressions for $S(a, m)$ and dividing by 2, we get

$$S(a, m) = 2m + \sum_{t=1}^{2m-1} 2m(-1)^t \cos^a \frac{t\pi}{m}$$

$$= 2m \sum_{t=0}^{2m-1} (-1)^t \cos^a \frac{t\pi}{m}$$

This sum vanishes when $a$ and $m$ have different parities, i.e. if $a$ is odd (recall that we assume $m$ is even). For even $a$, we have

$$S(a, m) = 4m \sum_{t=0}^{m-1} (-1)^t \cos^a \frac{t\pi}{m}.$$

This is a trigonometric power sum with known closed form expressions. In particular, Equation 3.4 from [DFGK17, §3] (after correcting a typo) tells us that

$$T(a,m) = \sum_{t=0}^{m-1} (-1)^t \cos^a \frac{t\pi}{m} = \begin{cases} 2^{1-a}m \left( \displaystyle\sum_{p=1}^{\lfloor a/m \rfloor} \binom{a}{a/2 - pm/2} - \sum_{p=1}^{\lfloor a/2m \rfloor} \binom{a}{a/2 - pm} \right) & a \geq 2m \\[2em] 2^{1-a}m \displaystyle\sum_{p=1}^{\lfloor a/m \rfloor} \binom{a}{a/2 - pm/2} & m \leq a < 2m \\[2em] 0 & a < m \end{cases}$$

$$= \begin{cases} 2^{1-a}m \left( \displaystyle\sum_{\substack{p=1 \\ p \text{ odd}}}^{\lfloor a/m \rfloor} \binom{a}{a/2 - pm/2} \right) & a \geq m \\[2em] 0 & a < m \end{cases}$$

To get a sense for the asymptotics as $a \to \infty$, we consider $a \gg m$ (say $a \geq 100m$). In this regime the sum of binomial coefficients in the sum above is seen to be $\Omega(2^a/\sqrt{a})$ (the $p = 1$ term alone contributes roughly $\binom{a}{a/2}$), and we get that $T(a,m) = \Omega(m/\sqrt{a})$.

This means $S(a,m) = 0$ for odd $a$ and $S(a,m) = 4mT(a,m) = \Omega(m^2/\sqrt{a})$ for large, even $a$. Substituting this back into Eq. (7), we get that

$$\|g\|^2 = \Omega \left( \sum_{\substack{a \gg m \\ a \text{ even}}} \frac{\widehat{\phi}_a^2}{\sqrt{a}} \right).$$

$\square$

We can now consider the special cases of $\phi = \text{ReLU}$ and $\phi = \sigma$ (the standard sigmoid) that are of interest.

**Corollary D.2.** *Consider $g$ instantiated with $\phi = \text{ReLU}$. Then $\|g\| = \Omega(1/m)$.*

*Proof.* The Hermite coefficients of ReLU satisfy $\widehat{\phi}_a = \Theta(a^{-5/4})$ (Lemma F.1). Thus by Lemma D.1,

$$\|g\|^2 = \Omega(\sum_{\substack{a \geq 100m \\ a \text{ even}}} a^{-3}) = \Omega(1/m^2).$$

$\square$

**Corollary D.3.** *Consider $g$ instantiated with $\phi = \sigma$, the standard sigmoid. Then $\|g\| = e^{-O(\sqrt{m})}$.*

*Proof.* The Hermite coefficients of $\sigma$ asymptotically satisfy $\widehat{\phi}_a \simeq e^{-C\sqrt{a}}$ [GGJ+20, §A.2] for some $C$. Thus by Lemma D.1,

$$\|g\|^2 = \Omega(\sum_{\substack{a \geq 100m \\ a \text{ even}}} \frac{e^{-\sqrt{a}}}{\sqrt{a}}).$$

The result then follows by the following standard integral approximation:

$$\sum_{t=N}^{\infty} \frac{e^{-\sqrt{t}}}{\sqrt{t}} \approx \int_N^{\infty} \frac{e^{-\sqrt{t}}}{\sqrt{t}} \, dt = 2e^{-\sqrt{N}}.$$

$\square$

We can now translate these into norm lower bounds on $f = \psi \circ g$. For us it suffices to consider $\psi = \tanh : \mathbb{R} \to [-1,1]$, which is essentially the sigmoid centered at 0. The centering at 0 and the output range being $[-1,1]$ is what is important to us, because we use $f$ to capture the conditional mean function of a $p$-concept.

**Lemma D.4.** *Consider $f$ instantiated with $\psi = \tanh$ and $\phi = \mathrm{ReLU}$. Then $\|f\| = \Omega(1/m^6)$.*

*Proof.* Ideally we would like to use the norm bound on $g$ to obtain an anti-concentration inequality of the form $\mathbb{P}[|g(x)| > t]$, and then translate that into a norm lower bound for $f$, but this is not immediate because $g$ is unbounded. So we introduce the function $g^T$, which is the same as $g$ except with the truncated ReLU, $\mathrm{ReLU}^T(x) = \min(T, \mathrm{ReLU}(x))$ ($T$ to be determined), in place of all standard ReLUs. Clearly $|g^T(x)| \le T$ for all $x$. It is also easy to see by a union bound that

$$\mathbb{P}[g(x) \ne g^T(x)] \le 2m \Pr_{t \sim \mathcal{N}(0,1)}[\mathrm{ReLU}(t) \ne \mathrm{ReLU}^T(t)] \le 2me^{-T^2/2},$$

since each $w_i$ is a unit vector.

Let $\mathrm{ReLU}_w(x)$ be shorthand for $\mathrm{ReLU}(x \cdot w)$, and similarly $\mathrm{ReLU}_w^T$. Observe first that

$$\|g - g^T\| = \frac{1}{2m} \left\| \sum_{i=1}^{2m} (-1)^i (\mathrm{ReLU}_{w_i} - \mathrm{ReLU}_{w_i}^T) \right\|$$

$$\le \frac{1}{2m} \sum_{i=1}^{2m} \|\mathrm{ReLU}_{w_i} - \mathrm{ReLU}_{w_i}^T)\|$$

$$= \|\mathrm{ReLU} - \mathrm{ReLU}^T\|_{\mathcal{N}(0,1)}$$

$$\le \sqrt{e^{-\frac{T^2}{2}} \left( T^2 + 1 - \frac{T}{\sqrt{2\pi}} \right)}$$

where the third equality again uses the fact the $w_i$ are unit vectors, and the last inequality is Lemma D.5. By picking $T = \Theta(m)$, this coupled with the fact that $\|g\| = \Omega(1/m)$ (Corollary D.2) tells us that $\|g^T\| = \Omega(1/m)$ as well.

This bound on $\|g^T\|$ yields an anti-concentration inequality for $g^T$ as follows:

$$\|g^T\|^2 = \mathbb{E}[g^T(x)^2] \le t^2\, \mathbb{P}[|g^T(x)| \le t] + T^2\, \mathbb{P}[|g^T(x)| > t] = t^2 + (T^2 - t^2)\, \mathbb{P}[|g^T(x)| > t],$$

so that

$$\mathbb{P}[|g^T(x)| > t] \ge \frac{\|g^T\|^2 - t^2}{T^2 - t^2}.$$

Recall that $\mathbb{P}[g(x) \ne g^T(x)] \le 2me^{-T^2/2}$, so

$$\mathbb{P}[|g(x)| > t] \ge \frac{\|g^T\|^2 - t^2}{T^2 - t^2} - 2me^{-T^2/2}.$$

Thus by taking $T = \Theta(m)$ and $t = \Theta(1/m)$, we get that

$$\mathbb{P}[|g(x)| > \Theta(1/m)] \ge \Omega(1/m^4).$$

Thus finally we have

$$\|f\| = \mathbb{E}[\tanh(g(x))^2] \ge \tanh^2(\Theta(1/m))\Omega(1/m^4) \ge \Omega(1/m^6),$$

since $\tanh(x) \approx x - x^3$ for small $x$ (by its Taylor series). $\qquad\square$

**Lemma D.5** ([GGJ+20], Appendix A.1)**.** *For $\mathrm{ReLU}^T(x) = \min(T, \mathrm{ReLU}(x))$,*

$$\|\mathrm{ReLU} - \mathrm{ReLU}^T\|_{\mathcal{N}(0,1)} \le \sqrt{e^{-\frac{T^2}{2}} \left( T^2 + 1 - \frac{T}{\sqrt{2\pi}} \right)}.$$

*Proof.* Let $p(t) = \frac{1}{\sqrt{2\pi}} e^{-t^2/2}$ be the pdf of $\mathcal{N}(0,1)$. Then

$$\|\mathrm{ReLU} - \mathrm{ReLU}^T\|^2_{\mathcal{N}(0,1)} = \mathbb{E}_{t \sim \mathcal{N}(0,1)}\left[ \left( \mathrm{ReLU}(t) - \mathrm{ReLU}^T(t) \right)^2 \right]$$

$$= \int_T^\infty (t - T)^2 p(t)\, dt$$

$$= \int_T^\infty t^2 p(t)\, dt - 2T \int_T^\infty t p(t)\, dt + T^2 \int_T^\infty p(t)\, dt$$

Noting that $p'(t) = -tp(t)$, we have

$$
\begin{aligned}
\int_T^\infty t^2 p(t)\, dt &= \int_T^\infty -t\, d(p(t)) \\
&= -t\, p(x)\Big|_T^\infty + \int_T^\infty p(t)\, dt & \text{(integration by parts)} \\
&= T\, p(T) + \Pr_{t \sim \mathcal{N}(0,1)}(t > T),
\end{aligned}
$$

$$
\int_T^\infty t\, p(t)\, dt = -p(t)\Big|_T^\infty = p(T),
$$

$$
\int_T^\infty p(t)\, dt = \Pr_{t \sim \mathcal{N}(0,1)}(t > T) \le e^{-\frac{T^2}{2}}.
$$

The claim follows by algebra. $\qquad\square$

**Lemma D.6.** *Consider $f$ instantiated with $\psi = \tanh$ and $\phi = \sigma$. Then $\|f\| = e^{-O(\sqrt{m})}$.*

*Proof.* Here the same approach as above becomes considerably simpler since $|g(x)| \le 1$ always. The norm bound on $g$ yields the following anti-concentration inequality:

$$
\mathbb{P}[|g(x)| > t] \ge \frac{\|g\|^2 - t^2}{1 - t^2}.
$$

In our case, taking $t = e^{-C\sqrt{m}}$ for sufficiently large $C$ and using $\|g\| = e^{-O(\sqrt{m})}$ (Corollary D.3) yields

$$
\mathbb{P}[|g(x)| > e^{-C\sqrt{m}}] = e^{-O(\sqrt{m})}.
$$

Thus

$$
\|f\| = \mathbb{E}[\tanh(g(x))^2] \ge \tanh^2(e^{-C\sqrt{m}}) e^{-O(\sqrt{m})} \ge e^{-O(\sqrt{m})},
$$

since again $\tanh(x) \approx x - x^3$ for small $x$. $\qquad\square$

# E  Agnostic learner for maximizing correlation

Here we will construct an SQ learner that can maximize correlation with ReLUs and sigmoids in the agnostic setting, i.e. satisfy Assumption 3.1.

Let $\mathcal{D}$ be a distribution on $\mathbb{R}^n \times \mathbb{R}$ such that the marginal on $\mathbb{R}^n$ is $\mathcal{N}(0, I_n)$. Let $f_{\mathsf{cmf}}(x) = \mathbb{E}[y|x]$ be the conditional mean function of $\mathcal{D}$, and assume $\|f_{\mathsf{cmf}}\| \le C$ for some constant $C$. Observe that for any $f$, the correlation $\mathbb{E}_{(x,y) \sim \mathcal{D}}[f(x)y]$ equals $\langle f, f_{\mathsf{cmf}} \rangle$. Let $\mathcal{H}$ be a hypothesis class with $\delta$-approximate degree $d$ ($\delta$ to be determined), and let $R = \operatorname{diam}(\mathcal{H})/2$. Let $h_{\mathsf{opt}} \in \mathcal{H}$ achieve $\max_{h \in \mathcal{H}} \langle h, f_{\mathsf{cmf}} \rangle$.

Our algorithm will be based on approximating the low-degree Hermite coefficients of $f_{\mathsf{cmf}}$, which is equivalent to performing polynomial $L^2$ regression. It is well-known that in this context, where $d$ is the $\delta$-approximate degree, polynomial $L^1$ regression up to degree $d$ gives a squared loss guarantee of $\delta$ [KKMS08]. But we will not be able to use this result directly since what we seek is a correlation guarantee. Instead, our approach will involve a sequence of inequalities relating the correlation achieved by $f_{\mathsf{cmf}}$, $h_{\mathsf{opt}}$, and their degree-$d$ approximations. A slight subtlety to keep in mind is that correlation can always be increased by scaling the function. This means that wherever scaling is possible, we have to take some care to rescale functions to have the maximum allowed norm, $R$.

Let $h_{\mathsf{opt}}^{\le d}$ and $f_{\mathsf{cmf}}^{\le d}$ be the Hermite components of degree at most $d$ of $h_{\mathsf{opt}}$ and $f_{\mathsf{cmf}}$ respectively. Let $\tilde{f}_{\mathsf{cmf}}^{\le d} = \frac{R}{\|f_{\mathsf{cmf}}^{\le d}\|} f_{\mathsf{cmf}}^{\le d}$. Among polynomials of degree $d$ in $\mathcal{B}(R)$, it is easy to see that $\tilde{f}_{\mathsf{cmf}}^{\le d}$ maximizes $\langle f, f_{\mathsf{cmf}} \rangle$, so that

$$
\langle \tilde{f}_{\mathsf{cmf}}^{\le d}, f_{\mathsf{cmf}} \rangle \ge \langle h_{\mathsf{opt}}^{\le d}, f_{\mathsf{cmf}} \rangle.
$$

Our agnostic learner will look to approximate $\tilde{f}_{\mathsf{cmf}}^{\le d}$ by outputting $p$ defined as follows. Suppose $f_{\mathsf{cmf}} = \sum_{I \in \mathbb{N}^n} \alpha_I H_I$, where $H_I$ is the multivariate Hermite polynomial of index $I$. For each $I$ of

total degree at most $d$, which we denote as $|I| \leq d$, let $\beta_I$ be our estimate of $\alpha_I = \langle f_{\mathsf{cmf}}, H_I \rangle$ to within tolerance $\tau$ (to be determined). This can be done using $n^{O(d)}$ queries of tolerance $\tau$. Let $\tilde{f} = \sum_{|I| \leq d} \beta_I H_I$, and finally let $p = \frac{R}{\|\tilde{f}\|} \tilde{f}$. We have

$$
\begin{aligned}
\|\tilde{f}_{\mathsf{cmf}}^{\leq d} - p\|^2 &= R^2 \left\| \frac{f_{\mathsf{cmf}}^{\leq d}}{\|f_{\mathsf{cmf}}^{\leq d}\|} - \frac{\tilde{f}}{\|\tilde{f}\|} \right\|^2 \\
&= R^2 \left\| \frac{f_{\mathsf{cmf}}^{\leq d} - \tilde{f}}{\|f_{\mathsf{cmf}}^{\leq d}\|} + \tilde{f} \left( \frac{1}{\|f_{\mathsf{cmf}}^{\leq d}\|} - \frac{1}{\|\tilde{f}\|} \right) \right\|^2 \\
&\leq 2R^2 \left( \frac{\|f_{\mathsf{cmf}}^{\leq d} - \tilde{f}\|^2}{\|f_{\mathsf{cmf}}^{\leq d}\|^2} + \|\tilde{f}\|^2 \left( \frac{1}{\|f_{\mathsf{cmf}}^{\leq d}\|} - \frac{1}{\|\tilde{f}\|} \right)^2 \right) \\
&= 2R^2 \left( \frac{\|f_{\mathsf{cmf}}^{\leq d} - \tilde{f}\|^2}{\|f_{\mathsf{cmf}}^{\leq d}\|^2} + \left( \frac{\|f_{\mathsf{cmf}}^{\leq d}\| - \|\tilde{f}\|}{\|f_{\mathsf{cmf}}^{\leq d}\|} \right)^2 \right) \\
&\leq 4R^2 \frac{\|f_{\mathsf{cmf}}^{\leq d} - \tilde{f}\|^2}{\|f_{\mathsf{cmf}}^{\leq d}\|^2} \qquad \qquad \text{(triangle ineq.)} \\
&\leq \frac{4R^2 n^d \tau^2}{\|f_{\mathsf{cmf}}^{\leq d}\|^2}, \qquad \qquad \qquad \qquad \qquad \qquad \text{(8)}
\end{aligned}
$$

since $\|\tilde{f} - f_{\mathsf{cmf}}^{\leq d}\| \leq n^{d/2} \tau$.

We claim that we can assume WLOG that $\|\tilde{f}_{\mathsf{cmf}}^{\leq d}\| \geq \epsilon/(2R)$. Indeed, we know $\max_{h \in \mathcal{H}} \langle h, f_{\mathsf{cmf}} \rangle = \langle h_{\mathsf{opt}}, f_{\mathsf{cmf}} \rangle$ and also $\|h_{\mathsf{opt}} - h_{\mathsf{opt}}^{\leq d}\| \leq \delta$. This implies that

$$
R \|\tilde{f}_{\mathsf{cmf}}^{\leq d}\| = \langle \tilde{f}_{\mathsf{cmf}}^{\leq d}, f_{\mathsf{cmf}} \rangle \geq \langle h_{\mathsf{opt}}^{\leq d}, f_{\mathsf{cmf}} \rangle \geq \langle h_{\mathsf{opt}}, f_{\mathsf{cmf}} \rangle - C\delta,
$$

where the last inequality is Cauchy–Schwarz. If $\langle h_{\mathsf{opt}}, f_{\mathsf{cmf}} \rangle \leq \epsilon$ then $0$ is a valid agnostic learner. Therefore, we can assume that $\langle h_{\mathsf{opt}}, f_{\mathsf{cmf}} \rangle \geq \epsilon$. Choosing $\delta = \frac{\epsilon}{2C}$, this means $\|\tilde{f}_{\mathsf{cmf}}^{\leq d}\| \geq \epsilon/(2R)$.

By Eq. (8), we then have

$$
\|\tilde{f}_{\mathsf{cmf}}^{\leq d} - p\| \leq \frac{4Rn^{d/2}\tau}{\epsilon}. \qquad \qquad \qquad \qquad \qquad \qquad \text{(9)}
$$

Now observe that

$$
\begin{aligned}
\langle p, f_{\mathsf{cmf}} \rangle &= \langle \tilde{f}_{\mathsf{cmf}}^{\leq d}, f_{\mathsf{cmf}} \rangle + \langle p - \tilde{f}_{\mathsf{cmf}}^{\leq d}, f_{\mathsf{cmf}} \rangle \\
&\geq \langle h_{\mathsf{opt}}^{\leq d}, f_{\mathsf{cmf}} \rangle - \frac{4RCn^{d/2}\tau}{\epsilon} \qquad \text{(Eq. (9) and Cauchy–Schwarz)} \\
&= \langle h_{\mathsf{opt}}, f_{\mathsf{cmf}} \rangle + \langle h_{\mathsf{opt}}^{\leq d} - h_{\mathsf{opt}}, f_{\mathsf{cmf}} \rangle - \frac{4RCn^{d/2}\tau}{\epsilon} \\
&\geq \langle h_{\mathsf{opt}}, f_{\mathsf{cmf}} \rangle - \frac{\epsilon}{2} - \frac{4RCn^{d/2}\tau}{\epsilon}. \qquad \text{(Cauchy–Schwarz, and using } \delta C = \epsilon/2)
\end{aligned}
$$

Setting $\tau = \frac{\epsilon^2}{8RCn^{d/2}}$ gives us the desired result.

## F  Approximate degree of ReLUs and sigmoids

Here we give estimates for the $\delta$-approximate degree of ReLUs and sigmoids under the standard Gaussian using bounds on their Hermite coefficients. Recall that we consider units $\phi(w \cdot x)$ with $\|w\|_2 \leq 1$. It is clear that for $\phi = \mathrm{ReLU}$ and $\phi = \sigma$, the norm only increases monotonically with $\|w\|_2$, so for the purposes of analysis it suffices to consider exactly $\|w\|_2 = 1$.

It is not hard to show that whenever $w$ is a unit vector, the total-degree-$d$ Hermite weight of $\phi(w \cdot x)$ as $x \sim \mathcal{N}(0, I_n)$ is the same as that of the univariate $\phi(t)$ as $t \sim \mathcal{N}(0, 1)$. (A quick way of seeing

this is to note that by rotational symmetry, we may assume WLOG that $w = e_1$, in which case the calculation is very straightforward.)

In what follows, we say $\widehat{\phi}_a$ are the Hermite coefficients of $\phi : \mathbb{R} \to \mathbb{R}$ if $\phi = \sum_a \widehat{\phi}_a H_a$, where the $H_a$ are the normalized probabilists' Hermite polynomials. We use $\tilde{H}_a$ to denote the un-normalized (i.e. monic) Hermite polynomials. (Note that this is somewhat nonstandard notation.)

First we consider ReLUs.

**Lemma F.1.** $\widehat{\text{ReLU}}_0 = 1/\sqrt{2\pi}$, $\widehat{\text{ReLU}}_1 = 1/2$ *and for* $a \geq 2$, $\widehat{\text{ReLU}}_a = \frac{1}{\sqrt{2\pi a!}}(\tilde{H}_a(0) + a\tilde{H}_{a-2}(0))$. *In particular,* $\widehat{\text{ReLU}}_a = 0$ *for odd* $a \geq 3$ *and* $|\widehat{\text{ReLU}}_a| = \Theta(a^{-5/4})$ *for even* $a$.

*Proof.* We use the following standard recurrence relation: $\tilde{H}_{a+1}(x) = x\tilde{H}_a(x) - a\tilde{H}_{a-1}(x)$. For $a \geq 2$,

$$\widehat{\text{ReLU}}_a = \frac{1}{\sqrt{2\pi}} \int_{-\infty}^{\infty} \text{ReLU}(x)H_a(x)e^{-\frac{x^2}{2}} dx$$

$$= \frac{1}{\sqrt{2\pi a!}} \int_0^{\infty} x\tilde{H}_a(x)e^{-\frac{x^2}{2}} dx$$

$$= \frac{1}{\sqrt{2\pi a!}} \int_0^{\infty} (\tilde{H}_{a+1}(x) + a\tilde{H}_{a-1}(x))e^{-\frac{x^2}{2}} dx$$

$$= \frac{1}{\sqrt{2\pi a!}}(\tilde{H}_a(0) + a\tilde{H}_{a-2}(0)).$$

Since $\tilde{H}_a(0) = 0$ for odd $a$, $\widehat{\text{ReLU}}_a = 0$ as well. For even $a = 2b$ with $b \geq 2$, by standard expressions for $\tilde{H}_a(0)$, we have

$$\widehat{\text{ReLU}}_a = \frac{1}{\sqrt{2\pi(2b)!}}(\tilde{H}_{2b}(0) + 2b\tilde{H}_{2b-2}(0))$$

$$= \frac{1}{\sqrt{2\pi(2b)!}} \left( (-1)^b \frac{(2b)!}{b!2^b} + 2b(-1)^{b-1} \frac{(2b-2)!}{(b-1)!2^{b-1}} \right)$$

$$= \frac{(-1)^b \sqrt{(2b)!}}{\sqrt{2\pi}b!2^b} \left( 1 - \frac{2b}{2b-1} \right)$$

$$= \frac{(-1)^{b+1} \sqrt{(2b)!}}{\sqrt{2\pi}(2b-1)b!2^b}$$

$$\approx \frac{(-1)^{b+1}}{\sqrt{2\pi}(2b-1)(2b)^{1/4}}$$

$$\approx \frac{(-1)^{b+1}}{b^{5/4}}$$

Here the second inequality follows from the fact $\binom{n}{n/2} \approx \frac{2^{n/2}}{\sqrt{n}}$. $\qquad\square$

**Corollary F.2.** *The $\delta$-approximate degree of* ReLU *under* $\mathcal{N}(0,1)$ *is* $O(\delta^{-4/3})$.

*Proof.* Let $p$ denote the the Hermite expansion of ReLU truncated at degree $d$. By the fact that $|\widehat{\text{ReLU}}_a| = \Theta(a^{-5/4})$ for even $a$ (and 0 for odd $a$), we see that

$$\|p - \text{ReLU}\|^2 = \sum_{a>d} \widehat{\text{ReLU}}_a^2$$

$$= \sum_{\substack{a>d \\ a \text{ even}}} \Theta(a^{-5/2})$$

$$= \Theta(d^{-3/2}).$$

For this to be at most $\delta^2$, we only need $d = O(\delta^{-4/3})$. $\qquad\square$

Now we turn to sigmoids. Let $\sigma$ denote the standard sigmoid, i.e. the logistic function $\sigma(t) = 1/(1 + e^{-t})$.

**Lemma F.3.** *For all sufficiently large $a$, $\widehat{\sigma}_a = e^{-\Omega(\sqrt{a})}$.*

*Proof.* Upper bounds on the Hermite coefficients of sigmoidal funtions are known to follow from classic results in the complex analysis of Hermite series [Hil40, Boy84]. We refer to [PSG19, Corollary F.7.1], where this computation is done for $\tanh'(x) = 1 - \tanh^2(x)$. The calculation is very similar for $\sigma$ (in fact, $\sigma$ is just an affine shift of $\tanh$). $\qquad\square$

**Corollary F.4.** *The $\delta$-approximate degree of $\sigma$ under $\mathcal{N}(0,1)$ is $\tilde{O}(\log^2 1/\delta)$.*

*Proof.* Let $p$ denote the Hermite expansion of $\sigma$ truncated at degree $d$. Observe that

$$\begin{aligned}
\|\sigma - p\|^2 &= \sum_{a > d} \widehat{\sigma}_a^2 \\
&= \sum_{a > d} e^{-\Omega(\sqrt{a})} \\
&= \Theta(\sqrt{d} e^{-\Omega(\sqrt{d})}),
\end{aligned}$$

which is at most $\delta^2$ for $d = \tilde{O}(\log^2 1/\delta)$. $\qquad\square$

## G   Frank–Wolfe convergence guarantee

Here we provide a self-contained proof of Theorem 2.5, restated here. In fact, we generalize the analysis to handle any constant factor approximation to the optimum, meaning that in the Frank–Wolfe subproblem of Algorithm 1, we only require

$$\langle s, -\nabla p(z_t)\rangle \geq \alpha \max_{s' \in \mathcal{Z}'} \langle s', -\nabla p(z_t)\rangle - \frac{1}{2}\delta\gamma_t C_p \tag{10}$$

for some constant $\alpha \leq 1$. We closely follow [Jag13, Appendix A], noting the differences in our slightly more general setup (the standard setup has $\mathcal{Z}' = \mathcal{Z}$, and $\alpha = 1$).

**Theorem G.1.** *Let $\mathcal{Z}' \subseteq \mathcal{Z}$ be convex sets, and let $p : \mathcal{Z} \to \mathbb{R}$ be a $\beta$-smoothly convex function. Let $C_p = \beta \operatorname{diam}(\mathcal{Z})^2$. Suppose that $z^* \in \mathcal{Z}'$ achieves $\min_{z' \in \mathcal{Z}'} p(z')$. For every $t$, the iterates of Algorithm 1 (modified to work with Eq. (10)) satisfy*

$$p(z_t) - p(z^*) \leq \frac{2C_p}{\alpha^2(t+2)}(1 + \delta).$$

*Proof.* Define the duality gap function $q : \mathcal{Z} \to \mathbb{R}$ as

$$q(z) = \max_{s \in \mathcal{Z}'} \langle z - s, \nabla p(z)\rangle.$$

Notice that $q$ takes in any $z \in \mathcal{Z}$ but maximizes only over $s \in \mathcal{Z}'$. By convexity of $p$ over $\mathcal{Z}$, we know that for all $z \in \mathcal{Z}, s \in \mathcal{Z}', p(z) + \langle s - z, \nabla p(z)\rangle \leq p(s)$, meaning that $p(z) - p(s) \leq q(z)$. In particular, $p(z) - p(z^*) \leq q(z)$, so that $q(z)$ always provides an upper bound on the gap between $p(z)$ and $p(z^*)$ — this is weak duality.

Next we establish the following guarantee on the progress made in each step, which corresponds to Lemma 5 in Jaggi's proof.

**Claim.** *Let the $t^{th}$ step be $z_{t+1} = z_t + \gamma(s - z_t)$, where $z_t, z_{t+1}, s \in \mathcal{Z}, \gamma \in [0, 1]$ is arbitrary, and $s$ satisfies*

$$\langle s, -\nabla p(z_t)\rangle \geq \alpha \max_{s' \in \mathcal{Z}'} \langle s', -\nabla p(z_t)\rangle - \frac{1}{2}\delta\gamma C_p.$$

*Then we have*

$$p(z_{t+1}) \leq p(z_t) - \alpha\gamma q(z_t) + \frac{\gamma^2}{2}C_p(1 + \delta).$$

To see this, first note that because $p$ is $\beta$-smoothly convex,

$$p(z_{t+1}) = p(z_t + \gamma(s - z_t))$$

$$\leq p(z_t) + \gamma\langle s - z_t, \nabla p(z_t)\rangle + \frac{\gamma^2}{2}C_p.$$

And from the way $s \in \mathcal{Z}$ was picked, we have

$$\langle s - z_t, -\nabla p(z_t)\rangle \geq \alpha \max_{s' \in \mathcal{Z}'}\langle s' - z_t, -\nabla p(z_t)\rangle - \frac{1}{2}\delta\gamma C_p$$

$$= \alpha q(z_t) - \frac{1}{2}\delta\gamma C_p.$$

The claim now follows.

As a consequence of the claim, we can say

$$p(z_{t+1}) - p(z^*) \leq p(z_t) - p(z^*) - \gamma q(z_t) + \frac{\gamma^2}{2}C_p(1 + \delta)$$

$$\leq (1 - \alpha\gamma)(p(z_t) - p(z^*)) + \frac{\gamma^2}{2}C_p(1 + \delta),$$

since $q(z_t) \geq p(z_t) - p(z^*)$ (weak duality). Taking $\gamma = \gamma_t = \frac{2}{\alpha(t+2)}$, the following bound can now by proven by induction on $t$:

$$p(z_t) - p(z^*) \leq \frac{2}{\alpha^2(t+2)}C_p(1 + \delta).$$

This proves the theorem. $\qquad\qquad\qquad\qquad\qquad\qquad\qquad\qquad\qquad\qquad\qquad\square$

## H  Smooth convexity of the loss functional

Let $\ell : \mathbb{R} \times \mathbb{R} \to \mathbb{R}$ be a loss function satisfying Assumption 3.2. Corresponding properties of the loss functional $L$ now follow. First we establish the subgradient, which will itself be an element of $L^2(\mathbb{R}^n, D)$, i.e. a function from $\mathbb{R}^n$ to $\mathbb{R}$. Let $f, h : \mathbb{R}^n \to \mathbb{R}$. Observe that at for every $x \in \mathbb{R}^n, y \in \mathbb{R}$, the subgradient property of $\ell$ tells us that

$$\ell(f(x) + h(x), y) - \ell(f(x), y) \geq \partial_1\ell(f(x), y)h(x).$$

Taking expectations over $(x, y) \sim D_{f^*}$, this yields

$$L(f + h) - L(f) \geq \mathop{\mathbb{E}}_{(x,y)\sim D_{f^*}}[\partial_1\ell(f(x), y)h(x)]$$

$$= \mathop{\mathbb{E}}_{x\sim D}[\mathop{\mathbb{E}}_{y|x}[\partial_1\ell(f(x), y)]h(x)]$$

$$= \langle s, h\rangle,$$

where

$$s : x \mapsto \mathop{\mathbb{E}}_{y|x}[\partial_1\ell(f(x), y)] = \mathop{\mathbb{E}}_{y\sim f^*(x)}[\partial_1\ell(f(x), y)]$$

is thus a subgradient of $L$ at $f$. $\beta$-smooth convexity is also easily established. Taking expectations over $(x, y) \sim D_{f^*}$ of

$$\ell(f(x) + h(x), y) - \ell(f(x), y) - \partial_1\ell(f(x), y)h(x) \leq \frac{\beta}{2}h(x)^2,$$

we get

$$L(f + h) - L(f) - \langle s, h\rangle \leq \frac{\beta}{2}\|h\|^2$$

for the same subgradient $s$. By smooth convexity, this subgradient is unique and so we can say that the gradient of $L$ at $f$ is given by $\nabla L(f) : x \mapsto \mathbb{E}_{y\sim f^*(x)}[\partial_1\ell(f(x), y)]$.

# I Relationship between Boolean 0-1 loss and real-valued correlation loss

Let $\mathcal{D}$ be a distribution on $\mathbb{R}^n \times \mathbb{R}$. Our lower bound applies against agnostic learners that satisfy Assumption 3.1, with a real-valued correlation guarantee, i.e. learners that learn a class $\mathcal{H}$ by outputting $f : \mathbb{R}^n \to \mathbb{R}$ such that

$$\mathbb{E}_{(x,y)\sim\mathcal{D}}[f(x)y] \geq \max_{g\in\mathcal{H}} \mathbb{E}_{(x,y)\sim\mathcal{D}}[g(x)y] - \epsilon. \tag{11}$$

In the Boolean setting, where the labels are $\{\pm 1\}$-valued, we have a distribution $P$ on $\mathbb{R}^n \times \{\pm 1\}$. A learner is said to agnostically learn $\mathcal{H}$ in terms of 0-1 loss if it is able to output $f : \mathbb{R}^n \to \{\pm 1\}$ such that

$$\mathbb{P}_{(a,b)\sim P}[f(a) \neq b] \leq \min_{g\in\mathcal{H}} \mathbb{P}_{(a,b)\sim P}[g(a) \neq b] + \epsilon,$$

or equivalently

$$\mathbb{E}_{(a,b)\sim P}[f(a)b] \geq \max_{g\in\mathcal{H}} \mathbb{P}_{(a,b)\sim P}[g(a)b] - \epsilon/2,$$

since $\mathbb{E}_{(a,b)\sim P}[f(a)b] = 1 - 2\,\mathbb{P}_{(a,b)\sim P}[f(a) \neq b]$. (The latter formulation has the benefit of making sense even for real-valued $f : \mathbb{R}^n \to \mathbb{R}$.)

It is not obvious that a learner $L$ of the above kind (with a Boolean 0-1 loss guarantee) gives us a real-valued correlation loss guarantee, because it only knows how to operate on distributions $P$ on $\mathbb{R}^n \times \{\pm 1\}$ (with Boolean labels), not distributions $\mathcal{D}$ on $\mathbb{R}^n \times \mathbb{R}$ (with arbitrary real labels). Moreover, in the SQ setting, we must be able to translate $L$'s queries to $P$, which are of the form $\phi : \mathbb{R}^n \times \{\pm 1\} \to \mathbb{R}$, into queries to $\mathcal{D}$. We claim that both of these difficulties can be gotten around. We will show that if $\mathcal{D}$ has bounded labels, say in $[-C, C]$, we can construct a distribution $P$ on $\mathbb{R}^n \times \{\pm 1\}$ and simulate $L$ on $P$ to obtain a correlation loss guarantee wrt $\mathcal{D}$.

Indeed, let $D$ denote the marginal of $\mathcal{D}$ on $\mathbb{R}^n$; for us, $D$ is always $\mathcal{N}(0, I_n)$. Then $P$ can be constructed simply as follows: draw $a \sim D$, and then randomly pick $b \in \{\pm 1\}$ such that $\mathbb{E}[b|a] = (\mathbb{E}_{(x,y)\sim\mathcal{D}}[y|x=a])/C$. (One could think of this as the "p-concept trick".) Equivalently, pick

$$b = \begin{cases} 1 & \text{with probability } \frac{1+(\mathbb{E}_{(x,y)\sim\mathcal{D}}[y|x=a])/C}{2} \\ -1 & \text{otherwise} \end{cases}$$

One can easily see that for any $f : \mathbb{R}^n \to \mathbb{R}$,

$$\mathbb{E}_{(a,b)\sim P}[f(a)b] = \frac{1}{C} \mathbb{E}_{(x,y)\sim\mathcal{D}}[f(x)y],$$

so that using $L$ to learn up to 0-1 error $\epsilon$ gives a correlation loss guarantee up to $C\epsilon/2$. It remains to show that we can indeed simulate $L$'s queries to $P$ using only SQ access to $\mathcal{D}$. For any query $\phi : \mathbb{R}^n \times \{\pm 1\} \to \mathbb{R}$, observe that (since the marginal of $P$ on $\mathbb{R}^n$ is also $D$)

$$\mathbb{E}_{(a,b)\sim P}[\phi(a,b)] = \mathbb{E}_{a\sim D}\left[\phi(a,1)\frac{1 + (\mathbb{E}_{(x,y)\sim\mathcal{D}}[y|x=a])/C}{2} + \phi(a,-1)\frac{1 - (\mathbb{E}_{(x,y)\sim\mathcal{D}}[y|x=a])/C}{2}\right]$$

$$= \frac{1}{2} \mathbb{E}_{a\sim D}[\phi(a,1) + \phi(a,-1)] + \frac{1}{2C} \mathbb{E}_{(x,y)\sim\mathcal{D}}[(\phi(x,1) - \phi(x,-1))y].$$

This expression can be computed using two statistical queries to $\mathcal{D}$ (or even just one, since we know the marginal $D$).

In our reduction (Theorem 4.1), we end up using the base learner on labeled distributions $\mathcal{D}$ where the labels correspond to the loss functional's gradient; when using surrogate loss, the label for $x$ is $\psi(f^*(x)) - \psi(f(x))$. We see that this is indeed bounded in $[-2, 2]$, since $\psi : \mathbb{R} \to [-1, 1]$. Recall that in solving the Frank–Wolfe subproblem we needed to worry about simulating SQ access to this $\mathcal{D}$ using only SQ access to the true $D_{\psi\circ f^*}$ (see Eq. (4) and surrounding discussion). Here we actually have a further layer: we need to simulate SQ access to $P$ using SQ access to $\mathcal{D}$, itself simulated using actual SQ access to $D_{\psi\circ f^*}$. But it is easily verified that by the argument just outlined, no trouble arises here, and that one can in fact also "directly" simulate $P$ using $D_{\psi\circ f^*}$ by the same argument as used for Eq. (4).

## J  Relationship between square loss and correlation loss for ReLUs

Let $\mathcal{D}$ be a distribution on $\mathbb{R}^n \times \mathbb{R}$, and assume the labels are bounded in $[-C, C]$. Our lower bounds apply to agnostic learners that satisfy Assumption 3.1, with a guarantee in terms of correlation, where the output hypothesis $f$ must satisfy

$$\mathbb{E}_{(x,y)\sim\mathcal{D}}[f(x)y] \geq \max_{g \in \mathcal{H}} \mathbb{E}_{(x,y)\sim\mathcal{D}}[g(x)y] - \epsilon.$$

But agnostic learning of real-valued functions is usually phrased in terms of square loss:

$$\mathbb{E}_{(x,y)\sim\mathcal{D}}[(f(x) - y)^2] \leq \min_{g \in \mathcal{H}} \mathbb{E}_{(x,y)\sim\mathcal{D}}[(g(x) - y)^2] + \epsilon'.$$

Here we show that for the class of ReLUs, $\mathcal{H} = \mathcal{H}_{\mathrm{ReLU}}$, an agnostic learner $L$ with a square loss guarantee can be used to satisfy Assumption 3.1. Fundamentally, this amounts to working out a geometric relationship between distances and projections in our function space, and much of the following argument can be viewed as a somewhat careful elaboration of what, in the familiar Euclidean setup, is more easily visualized.

For simplicity, throughout this section we will scale the class $\mathcal{H}_{\mathrm{ReLU}}$ so that the maximum norm of any function is 1:

$$\mathcal{H} = \mathcal{H}_{\mathrm{ReLU}} = \{\pm\sqrt{2}\,\mathrm{ReLU}(u \cdot x) \mid \|u\|_2 \leq 1\}.$$

An important property of this class is that we can always scale a function $h \in \mathcal{H}$ to have any desired norm in $[0, 1]$ without leaving the class. That is, for any nonzero $h \in \mathcal{H}$ and any $\lambda \in [0, 1]$, $\frac{\lambda}{\|h\|}h \in \mathcal{H}$. This follows simply from the fact that $\|\mathrm{ReLU}(u \cdot x)\| = \|u\|_2/\sqrt{2}$. We can think of this as saying that $\mathcal{H}$ is a norm-bounded section of a convex cone.

Let $f_{\mathsf{cmf}}(x) = \mathbb{E}[y|x]$. Let $h_{\mathsf{sq}}$ be a minimizer over all $h \in \mathcal{H}$ of the squared loss, $\mathbb{E}_{(x,y)\sim\mathcal{D}}[(h(x) - y)^2]$. An equivalent and more convenient view is that this is a minimizer of the squared distance $\|h - f_{\mathsf{cmf}}\|^2$, since

$$\|h - f_{\mathsf{cmf}}\|^2 = \|h\|^2 - 2\langle h, f_{\mathsf{cmf}}\rangle + \|f_{\mathsf{cmf}}\|^2 = \mathbb{E}_{\mathcal{D}}[(h(x) - y)^2] + \|f_{\mathsf{cmf}}\|^2 - \mathbb{E}_{\mathcal{D}}[y^2],$$

and the latter terms are independent of $h$. This view is particularly important since it, combined with the fact that $\mathcal{H}$ is essentially a bounded convex cone, gives us an orthogonal projection theorem. Specifically, it is the case that the norm of $h_{\mathsf{sq}}$ must be the length of the projection of $f_{\mathsf{cmf}}$ onto the line $\lambda h_{\mathsf{sq}}$ for $\lambda \in [0, 1]$ (assuming this length is at most 1; otherwise, the norm is 1). In other words,

$$\|h_{\mathsf{sq}}\| = \min\{\langle \frac{h_{\mathsf{sq}}}{\|h_{\mathsf{sq}}\|}, f_{\mathsf{cmf}}\rangle, 1\}. \tag{12}$$

This can be seen by asking: for what $\lambda \in [0, 1]$ is $\|\frac{\lambda}{\|h_{\mathsf{sq}}\|}h_{\mathsf{sq}} - f_{\mathsf{cmf}}\|$ minimized? (The point being that $h_{\mathsf{sq}}$ could be rescaled to have norm $\lambda$.) By writing this as

$$\|\frac{\lambda}{\|h_{\mathsf{sq}}\|}h_{\mathsf{sq}} - f_{\mathsf{cmf}}\|^2 = \left(\lambda - \langle\frac{h_{\mathsf{sq}}}{\|h_{\mathsf{sq}}\|}, f_{\mathsf{cmf}}\rangle\right)^2 + \|f_{\mathsf{cmf}}\|^2 - \langle\frac{h_{\mathsf{sq}}}{\|h_{\mathsf{sq}}\|}, f_{\mathsf{cmf}}\rangle^2,$$

the observation follows immediately.[2] This projection theorem also tells us that $h_{\mathsf{sq}} = 0$ iff $f_{\mathsf{cmf}}$ has no projection onto any $h \in \mathcal{H}$, i.e. $\langle h, f_{\mathsf{cmf}}\rangle = 0$ for all $h \in \mathcal{H}$.[3]

Let $h_{\mathsf{cor}}$ be a maximizer of the correlation, $\mathbb{E}_{(x,y)\sim\mathcal{D}}[h(x)y] = \langle h, f_{\mathsf{cmf}}\rangle$. We may clearly assume that $h_{\mathsf{cor}}$ has the maximum possible norm, which is 1. We claim that in fact, $h_{\mathsf{cor}}$ can be taken to be $h_{\mathsf{sq}}/\|h_{\mathsf{sq}}\|$ (assuming $h_{\mathsf{sq}} \neq 0$; otherwise, $h_{\mathsf{cor}} = 0$ as well since, as noted, this means $\langle h, f_{\mathsf{cmf}}\rangle = 0$ for all $h \in \mathcal{H}$). To see why, first assume $h_{\mathsf{sq}} \neq 0$ and use the fact that for any nonzero $h \in \mathcal{H}$, the square loss achieved by $\frac{\|h_{\mathsf{sq}}\|}{\|h\|}h$ (i.e. $h$ scaled to have $h_{\mathsf{sq}}$'s norm) cannot be better than that of $h_{\mathsf{sq}}$ itself. Thus by an algebraic manipulation we have

$$\|h_{\mathsf{sq}} - f_{\mathsf{cmf}}\|^2 \leq \left\|\frac{\|h_{\mathsf{sq}}\|}{\|h\|}h - f_{\mathsf{cmf}}\right\|^2$$

$$\implies \langle\frac{h_{\mathsf{sq}}}{\|h_{\mathsf{sq}}\|}, f_{\mathsf{cmf}}\rangle \geq \langle\frac{h}{\|h\|}, f_{\mathsf{cmf}}\rangle \geq \langle h, f_{\mathsf{cmf}}\rangle.$$

Since this holds for any $h \in \mathcal{H}$, we may take $h_{\mathsf{cor}} = h_{\mathsf{sq}}/\|h_{\mathsf{sq}}\|$.

Now suppose we have an agnostic learner in terms of square loss that returns $h$ such that
$$\|h - f_{\mathsf{cmf}}\|^2 \leq \|h_{\mathsf{sq}} - f_{\mathsf{cmf}}\|^2 + \epsilon'.$$
For a suitable choice of $\epsilon'$ (depending on the final desired $\epsilon$), we would like to say that $h/\|h\|$ achieves correlation that is $\epsilon$-competitive with $h_{\mathsf{cor}}$. Indeed, if $h_{\mathsf{sq}} = 0$ this is trivial, since as noted this means $\langle h, f_{\mathsf{cmf}} \rangle = 0$ for all $h \in \mathcal{H}$. Otherwise, by comparing $\frac{\|h\|}{\|h_{\mathsf{sq}}\|} h_{\mathsf{sq}}$ (i.e. $h_{\mathsf{sq}}$ scaled to have $h$'s norm) with $h_{\mathsf{sq}}$ itself, we may say that
$$\|h - f_{\mathsf{cmf}}\|^2 \leq \|h_{\mathsf{sq}} - f_{\mathsf{cmf}}\|^2 + \epsilon' \leq \left\| \frac{\|h\|}{\|h_{\mathsf{sq}}\|} h_{\mathsf{sq}} - f_{\mathsf{cmf}} \right\|^2 + \epsilon'.$$

Some rearrangement gives
$$\langle \frac{h}{\|h\|}, f_{\mathsf{cmf}} \rangle \geq \langle \frac{h_{\mathsf{sq}}}{\|h_{\mathsf{sq}}\|}, f_{\mathsf{cmf}} \rangle - \frac{\epsilon'}{2\|h\|}$$
$$= \langle h_{\mathsf{cor}}, f_{\mathsf{cmf}} \rangle - \frac{\epsilon'}{2\|h\|}, \tag{13}$$
showing that $h/\|h\|$ is $\frac{\epsilon'}{2\|h\|}$-competitive with $h_{\mathsf{cor}}$.

But an issue here is that $\|h\|$ could be very small, or even zero. We claim that we can actually address this separately as an easy case: it implies that we are in a trivial situation in which even the 0 function performs fairly well, and so even the best possible correlation must be quite small.

**Lemma J.1.** *Let $h$ be such that $\|h - f_{\mathsf{cmf}}\|^2 \leq \|h_{\mathsf{sq}} - f_{\mathsf{cmf}}\|^2 + \epsilon'$. Suppose $\|h\| \leq \eta$. Then $\langle h_{\mathsf{cor}}, f_{\mathsf{cmf}} \rangle \leq \sqrt{\epsilon' + 2C\eta}$. In particular, the 0 function is $\sqrt{\epsilon' + 2C\eta}$-competitive with $h_{\mathsf{cor}}$.*

*Proof.* By Cauchy–Schwarz,
$$\|0 - f_{\mathsf{cmf}}\|^2 - \|h - f_{\mathsf{cmf}}\|^2 = 2\langle h, f_{\mathsf{cmf}} \rangle - \|f_{\mathsf{cmf}}\|^2 \leq 2\|h\|\|f_{\mathsf{cmf}}\| \leq 2C\eta,$$
where we use $\|f_{\mathsf{cmf}}\| \leq C$ since the labels are assumed to be bounded in $[-C, C]$. Thus
$$\|0 - f_{\mathsf{cmf}}\|^2 \leq \|h - f_{\mathsf{cmf}}\|^2 + 2C\eta \leq \|h_{\mathsf{sq}} - f_{\mathsf{cmf}}\|^2 + \epsilon' + 2C\eta.$$
On the other hand, by definition of $h_{\mathsf{sq}}$,
$$\|h_{\mathsf{sq}} - f_{\mathsf{cmf}}\|^2 \leq \|0 - f_{\mathsf{cmf}}\|^2,$$
Put together, this means that the 0 function achieves nearly the same square loss as $h_{\mathsf{sq}}$:
$$\|h_{\mathsf{sq}} - f_{\mathsf{cmf}}\|^2 \leq \|0 - f_{\mathsf{cmf}}\|^2 \leq \|h_{\mathsf{sq}} - f_{\mathsf{cmf}}\|^2 + \epsilon' + 2C\eta. \tag{14}$$
This lets us conclude that $\|h_{\mathsf{sq}}\|$ must be small:
$$\|h_{\mathsf{sq}}\|^2 = \|f_{\mathsf{cmf}}\|^2 - \|h_{\mathsf{sq}} - f_{\mathsf{cmf}}\|^2 + 2\langle h_{\mathsf{sq}} - f_{\mathsf{cmf}}, h_{\mathsf{sq}} \rangle \leq \epsilon' + 2C\eta,$$
where we use Eq. (14) and the fact that by can rewrite Eq. (12) as $\|h_{\mathsf{sq}}\| \leq \langle \frac{h_{\mathsf{sq}}}{\|h_{\mathsf{sq}}\|}, f_{\mathsf{cmf}} \rangle$, or $\langle h_{\mathsf{sq}} - f_{\mathsf{cmf}}, h_{\mathsf{sq}} \rangle \leq 0$. But now since $\|h_{\mathsf{sq}}\| \leq \sqrt{\epsilon' + 2C\eta} < 1$ ($\epsilon'$ and $\eta$ will be picked sufficiently small), Eq. (12) boils down to saying that
$$\langle h_{\mathsf{cor}}, f_{\mathsf{cmf}} \rangle = \langle \frac{h_{\mathsf{sq}}}{\|h_{\mathsf{sq}}\|}, f_{\mathsf{cmf}} \rangle = \|h_{\mathsf{sq}}\| \leq \sqrt{\epsilon' + 2C\eta}.$$
$\square$

We can now put everything together.

**Theorem J.2.** *Suppose we have an agnostic learner $L$ for $\mathcal{H}_{\mathrm{ReLU}}$ under $\mathcal{D}$ with a square loss guarantee. Then $L$ can be used to yield a correlation guarantee, i.e. to satisfy Assumption 3.1.*

*Proof.* Run $L$ with $\epsilon' = \Theta(\epsilon^3)$ to get $h$ such that $\|h - f_{\mathsf{cmf}}\|^2 \leq \|h_{\mathsf{sq}} - f_{\mathsf{cmf}}\|^2 + \epsilon'$. By Lemma J.1, if $\|h\| \leq \eta = \Theta(\epsilon^2)$, then 0 is $\epsilon$-competitive with $h_{\mathsf{cor}}$. So we may assume that $\|h\| \geq \Theta(\epsilon^2)$. But then by Eq. (13), since now $\frac{\epsilon'}{2\|h\|} \leq \epsilon$, we get that $h/\|h\|$ is $\epsilon$-competitive with $h_{\mathsf{cor}}$. $\square$