[Reviews · NeurIPS 2020]

Review 1

Summary and Contributions: This paper studies a basic problem in learning theory, namely that of *agnostically* learning the class of single neurons, for various activations (such as ReLU, sigmoid). In particular, the paper considers the "statistical query (SQ) model", which is very well studied in literature and most of the learning algorithms studied in literature can be implemented in the SQ model as well (with few notable exceptions of course). The main result in the paper is to prove limitations on SQ algorithms for these learning problems. The paper shows that *agnostically* learning ReLU/sigmoids/halfspaces to "correlational error" OPT + eps, requires essentially exp(n) many SQ queries as long as the tolerance of the queries is "very small". More than just the particular results, the technique used is quite novel. In particular, it is shown that any algorithm for agnostically learning a single neuron can be used to learn one-hidden layer neural networks in the probabilistic concept model (in the realizable setting). While such techniques have been used before, the novel aspect is to accomplish this within the SQ model, which the paper achieves by using a convex surrogate loss (used in prior works on learning generalized linear models). RESPONSE TO AUTHORS: Thank you for the response. My rating for the paper is unchanged, and I continue to feel the paper should be accepted.

Strengths: The paper identifies key challenges/barriers in obtained efficient algorithms for agnostically learning single neurons. These results will inform the future development of learning algorithms for these problems. The SQ implementation of gradient descent in the function space using Frank-Wolfe via the surrogate loss is novel and might on its own influence future work on learning algorithms more broadly.

Weaknesses: While the results and techniques are interesting, I find the trade-off between _number of SQ queries_ and _query tolerance_ to be a bit strange. In particular, the lower bound on SQ queries *decreases* when epsilon gets smaller. Of course, I understand that the tolerance is also getting smaller, so this is not a contradiction. But perhaps it will be more helpful to explictly chart out the tradeoff between #queries and tolerance in the lower bound. [I suppose, this might require digging into the details of the prior work on SQ lower bounds for one-hidden-layer neural nets.]

Correctness: The proofs seem correct to me at a high level.

Clarity: The paper is well written and the proofs are modular and clear.

Relation to Prior Work: The relation to prior work is adequately discussed.

Reproducibility: Yes

Additional Feedback: Line 32-34: It might help to make this sentence a bit more precise. For the squared loss, I suppose the above equation implies an OPT + 2*eps agnostic learning w.r.t. squared loss when ||h|| \ge c for all h \in H and we are restricted to ||f|| \le c. It will help to make this clear at least in the Appendix. Alg 1: C_p is not defined (I saw that it is defined in Appendix G, but will be good to define in the main body as well). Line 242: f(x) should be psi(f(x)) in the second line of the equation.


Review 2

Summary and Contributions: The work provides novel statistical-query lower bounds on learning single neurons with different non-linear activations, in the agnostic setting, when the underlying distribution is Gaussian. Specifically, the authors show new results on the hardness of learning a single-neuron with ReLU and sigmoid activation, and on learning half-spaces. The technique for showing these lower bounds is by reduction from the hardness of learning neural networks with probabilistic output from statistical-queries, which was recently shown in a previous work. To achieve this reduction, the authors show that an agnostic learner of single neurons can be used to learn a neural network, using the conditional gradient-descent algorithm, in function space, over the convex hull of the single-neuron functions. This shows that agnostic SQ learners for neurons can be boosted to learn probabilistic neural networks, using only statistical-queries, which are in turn hard to learn in the SQ model.

Strengths: This is a solid work, providing novel fundamental hardness result in the seemingly simple setting of agnostically learning a single neuron (or half-space) under the Gaussian distribution. This work completes the picture of the learnability of single neurons, which has been the focus of various recent works. The technique used for showing the hardness results is of independent interest, and sheds new light on the connection between agnostic learning of single neurons and learnability of two-layer networks. Small typo: in line 242, after the second equality, f(x) should be psi(f(x))

Weaknesses: -

Correctness: seems correct

Clarity: clear

Relation to Prior Work: yes

Reproducibility: Yes

Additional Feedback:


Review 3

Summary and Contributions: The statistical query (SQ) framework is a powerful algorithmic model that has been fruitful for understanding algorithmic limits for learning problems. This paper builds on recent work of Goel et al. (ICML 2020) and of Diakonikolas et al. (COLT 2020) giving improved lower bounds for SQ learning single-hidden-layer neural networks. The authors give strong lower bounds against agnostically learning a single ReLU in this setting.

Strengths: This paper gives quite strong (exponential) lower bounds for general SQ agnostic learning of a single ReLU. There has been a sequence of works giving SQ lower bounds in this area, and this is arguably the strongest and most general. The proof technique is a somewhat novel SQ reduction.

Weaknesses: The direct technical contribution of this paper is not insignificant, but it relies heavily on the recent Goel et al. and Diakonikolas et al. papers, which slightly limits its novelty.

Correctness: Yes. I did not check the appendices carefully, but the claims proved there seem innocuous.

Clarity: Excellent, clear writing with plenty of motivation, high level explanation of arguments and detailed proofs. My only complaint is that I had to dig into the Goel et al. paper to understand why the claim about lower bounds against general (as opposed to correlational) queries was justified.

Relation to Prior Work: The discussion on page 3 is adequate

Reproducibility: Yes

Additional Feedback: I have no new concerns about this paper after reading the author feedback.


Review 4

Summary and Contributions: The paper shows statistical query (SQ) lower bounds for ReLU, sigmoid and half-space regression. This is done via reduction to recently proved SQ lower bounds for learning one-hidden layer neural networks. This is a natural idea and has precedents in previous work in different but related contexts. The present context seems new and new technical ideas are needed to carry out the reduction. ---Post Rebuttal--- I am increasing the score. I am not satisfied with authors' response for the motivation for studying the problem of training single neurons. I am increasing on the basis of the results on threshold. The authors should check if they can achieve similar results for logistic regression and other generalized linear models. If that's true, that will provide motivation from classical ML. The paper needs to be much better written as discussed in the review.

Strengths: The reduction, carried out in the SQ model using convex optimization, could be useful for other problems.

Weaknesses: (1) I find the problem of learning ReLU and sigmoid artificial. I did see that there have been a bunch of papers on it in last couple of years, yet as far as I can see (by a quick look at some of these) there's no motivation beyond calling it a fundamental primitive and citing other works that have studied the problem. In particular, it's not clear what results about a single ReLU/sigmoid tell us about neural networks which is the problem of real interest. An answer to this is needed to justify calling ReLU regression a fundamental primitive. The present paper does the reverse: it draws conclusions about single ReLU from results about neural networks. The fact that there is a line of research about this problem cannot be a justification. Any clarification in this regard would be useful. I recognize that not all theoretical problems need have utilitarian motivation: aesthetic considerations can suffice. Also, one could go by presumed future utility of the results. I am skeptical on both counts but would like to know the authors' thoughts. Half-spaces, on the other hand, are natural and are of course very well-studied. But see the comments about this in the next item. (2) Theorems 1.1, 1.2, 1.3 need explanation and need to be put in context. E.g., what is the class of functions that can be output? Unpack the technical content of the theorem: what is the significance of the tolerance parameter \tau taking on a particular value given in the theorem? How does it relate to existing algorithms that are used? If space is a concern, this can be done in the appendix with a one-two sentence summary in the main paper. In particular, for Theorem 1.3, it would be useful to compare it with the upper bounds of [KKMS08, DKN10]. At first sight, they appear to contradict each other: take \epsilon to be a small positive constant, then the theorem shows an exponential SQ lower bound but there is a poly-time poly-query algorithm by [KKMS08, DKN10]. The answer presumably lies in the constant hidden in \Theta in the expression for \tau (which could make \tau too large)? Or is it that the algorithms in [KKMS08, DKN10] do not fall under the SQ model? (3) The paper is not easy to read for a reader not already familiar with the background (I work in ML theory, though had little familiarity with the SQ literature). E.g., Corollary 2.4 is hard to understand. What does "accompanying norm lower bound" mean? And how does it follow from the previous two results on SDA? What general theorem is being used? An outline of the proof that explained what each lemma was doing would be useful. As it is presently, I find the proof hard to follow and took me considerably longer than would have with more helpful writing. For example, at first reading, it appears that Theorem 5.1 is a conditional statement depending on the truth of Assumption 3.1. It would also be useful to say where in the paper Theorem 1.1 is being proved.

Correctness: The paper seems to be correct though I didn't verify every single detail.

Clarity: The writing is uneven with some parts well-written but a large part of the paper needs significant improvement.

Relation to Prior Work: Well done.

Reproducibility: Yes

Additional Feedback: The notion of concept class used in Def. 2.1 is not defined. This is presumably a real-valued concept class going by a previous paper with a definition of SDA? D_{f*} is not uniquely determined by f*. This makes some of the statements in the paper imprecise (e.g. the one on line 133) and I think they need to be amended with a quantification that they hold for all D_{f*} that can arise from a given f*. Give some intuition as to why use Frank--Wolfe as opposed to some other convex optimization method? Statistical dimension is given a technical definition; if it has an intuitive interpretation, that could be useful (put in the appendix if necessary).

[Author Response · NeurIPS 2020]

We thank all the reviewers for their comments.

**Reviewer #1.** Regarding the relationship between the query lower bound, the tolerance $\tau$ and the error $\epsilon$, note first
that we must have $\tau \lesssim \epsilon \lesssim \beta$ (where $\beta$ is a norm lower bound for the concept class), as we discuss in Appendix A.
Moreover, because of technical requirements on $\tau$ for the DKKZ20 result, we end up picking $\tau$ as a function of $\epsilon$ in our
reduction. So as $\epsilon$ decreases, $\tau$ decreases as well, and we get a series of incomparable (though still exponential) bounds
due to the tradeoffs between query complexity and tolerance. We will include a clearer discussion of these issues in
Appendix A. We will also address your additional feedback (including expanding on lines 32-34) when revising the
manuscript.

**Reviewer #4.** (1) A major goal in deep learning is to find provably efficient algorithms for learning classes of neural
networks. There are several papers a year on this topic. Here we are showing for the first time that if there is noise in
the labels, this goal is impossible even for the simplest networks, ones that consist of a single activation. As we mention
in the paper, our results rule out polynomial-time algorithms for *any* nonpolynomial activation (halfspaces and ReLU
are just examples). The activation can take Boolean or real-valued outputs. Since algorithms for learning polynomial
activations are known, this characterizes the computational complexity of learning single activations.

(2) Regarding the context for Theorems 1, 2, 3, we note that these lower bounds hold broadly for all statistical query
algorithms and hold regardless of the structure of the learner's output hypothesis. As for the tolerance, much of its
significance lies in capturing what in traditional PAC algorithms would be the sample complexity. Specifically, it takes
$\Theta(1/\tau^2)$ samples to simulate an SQ of tolerance $\tau$, and this is sometimes known as the *estimation complexity* of an SQ
algorithm.

As for a comparison with prior results such as KKMS08 and DKN10 for halfspaces, our SQ lower bound (like all SQ
lower bounds) states that any SQ algorithm must use *either* $\exp(n)$ queries *or* very small ($n^{-1/\epsilon}$) tolerance (which
corresponds to sample complexity). KKMS08 (with its sample complexity of $n^{1/\epsilon^2}$) falls into the latter category, and
our tolerance bounds show that this is nearly optimal.

(3) Regarding your comments on the difficulty of reading the paper without prior background in the SQ literature, we
acknowledge some of your points on where the writing could be improved. It is true that Corollary 2.4 is somewhat
confusing. The "accompanying norm lower bound" refers to a $\beta$ such that $\|g\| \geq \beta$ for all $g \in \mathcal{G}$, as used in Theorem
2.2. Part (a) of the corollary ((b) and (c) are similar) follows from "instantiating" the generic construction $\mathcal{G}$ of Theorem
2.3 with $\phi = \text{ReLU}$ to obtain say $\mathcal{G}_{\text{ReLU}}$, noting that functions in $\mathcal{G}_{\text{ReLU}}$ satisfy a norm lower bound of $\beta = \Omega(1/k^2)$
(with proofs deferred to Appendix D), and then using $\mathcal{C} = \mathcal{G}_{\text{ReLU}}$ in Theorem 2.2 to obtain the final lower bound. As
for Theorem 5.1, we do state "suppose that Assumption 3.1 holds for $\mathcal{H}_{\text{ReLU}}$" to try to be as clear as possible. As
for Theorem 1.1 (and 1.2 and 1.3), they are proved in Section 5, and we will note this. We will also try to make the
organization of the lemmas clearer.

Regarding additional feedback:

• The concept class in Def 2.1 can be either real-valued or Boolean.
• As defined in lines 124-126, $D_{f^*}$ in this case refers to the unique distribution on $\mathbb{R}^n \times \{\pm 1\}$ defined by the
(Boolean) $p$-concept $f^*$.
• One important reason for using Frank–Wolfe is because it avoids a projection step, as would be required in
say standard projected GD. In our $L^2$ function space, it is not natural to require the base learner to find such
a projection of the functional gradient onto $\text{conv}(\mathcal{H})$. Another important reason is that Frank–Wolfe uses a
*linear* optimization subproblem, and this is important in order to preserve "SQness" during the reduction (see
lines 215-227 and 242-245).

We will address all these points when revising the manuscript.

[Meta-Review · NeurIPS 2020]

The paper provides strong new lower bounds to an actively studies problem. Three of the reviewers are clearly recommending accepting the paper. Although one of the reviewers has some reservations for the motivation of this line of work, after discussion he's also leaning towards accepting the paper. Added after decisions: During the review process, we noticed that the problem and results in this submission are closely related to those in another submission. After I had written my original metareview, it was further brought to my attention that both of these submissions are on arXiv (the other one is arXiv:2006.16200) and the arXiv versions already acknowledge the parallel work. Now that both papers have been accepted to NeurIPS, I'm asking both sets of authors to include in their final versions a discussion explicitly comparing their results to those in the other paper.